# Mixed Effectiveness of REDD+ Subnational Initiatives after 10 Years of Interventions on the Yucatan Peninsula, Mexico

**Edward A. Ellis** [1,*][ID]**, José Antonio Sierra-Huelsz** [2]**, Gustavo Celestino Ortíz Ceballos** [3][ID]**, Citlalli López Binnqüist** [1] **and Carlos R. Cerdán** [3]

[1]  Centro de Investigaciones Tropicales, Universidad Veracruzana, Xalapa, Veracruz 91000, Mexico; cilopez@uv.mx
[2]  People and Plants International, Bristol, VT 05443, USA; jashpat@gmail.com
[3]  Facultad de Ciencias Agrícolas, Universidad Veracruzana, Xalapa, Veracruz 91000, Mexico; gusortiz@uv.mx (G.C.O.C.); ccerdan@uv.mx (C.R.C.)
[*]  Correspondence: ellis_eddie@yahoo.com

**Abstract:** Since 2010, the Reducing Emissions from Deforestation and Degradation (REDD+) mechanism has been implemented in Mexico's Yucatan Peninsula, a biodiversity hotspot with persistent deforestation problems. We apply the before-after-control-intervention approach and quasi-experimental methods to evaluate the effectiveness of REDD+ interventions in reducing deforestation at municipal (meso) and community (micro) scales. Difference-in-differences regression and propensity score matching did not show an overall reduction in forest cover loss from REDD+ projects at both scales. However, Synthetic Control Method (SCM) analyses demonstrated mixed REDD+ effectiveness among intervened municipalities and communities. Funding agencies and number of REDD+ projects intervening in a municipality or community did not appear to affect REDD+ outcomes. However, cattle production and commercial agriculture land uses tended to impede REDD+ effectiveness. Cases of communities with important forestry enterprises exemplified reduced forest cover loss but not when cattle production was present. Communities and municipalities with negative REDD+ outcomes were notable along the southern region bordering Guatemala and Belize, a remote forest frontier fraught with illegal activities and socio-environmental conflicts. We hypothesize that strengthening community governance and organizational capacity results in REDD+ effectiveness. The observed successes and problems in intervened communities deserve closer examination for REDD+ future planning and development of strategies on the Yucatan Peninsula.

**Keywords:** REDD+; Yucatan Peninsula; deforestation; before-after-control intervention; quasi-experimental methods; propensity score; matching; synthetic control method

## 1. Introduction

### 1.1. REDD+ on the Yucatan Peninsula: Time for Evaluation

In 2007, the global community adopted REDD+ (Reducing Emissions from Deforestation and Degradation) as a climate mitigation policy mechanism [1]. REDD+ raised both expectations and debate as it promised to be a key tool for participating countries to transition towards a carbon-neutral economy, focusing on subnational forest conservation [2] and rural smallholder farmer development interventions [3,4]. After more than 10 years since its inception, the need to evaluate the effectiveness of REDD+ in reducing deforestation has arrived. As a global endeavor, REDD+ has been extensively promoted across social and ecologically contrasting regions (mostly tropical) and has been implemented

using diverse policy and project strategies [3,5]. The first existing assessments suggest that REDD+ effectiveness varies between forest regions [6,7], which raises the need to analyze individual regions with closer detail and at multiple scales (i.e., local and landscape) [8].

As REDD+ passed from early stages to being implemented in 47 tropical and subtropical nations [9], it became one of the largest environmental policy interventions and an important field for testing analytical tools [10,11]. In a global review of REDD+ pilot projects, based on the development of an extensive database, Caplow et al. [12] found that the vast majority lacked proper impact evaluations and stressed the need for rigorous evaluation methods of current REDD+ projects. Experimental and quasi-experimental designs are favored as counterfactual outcomes are increasingly considered the gold standard against which policy interventions are to be gauged [8]. The Before-After-Control-Intervention (BACI) was regarded as a quasi-experimental approach that can best control both for the intervention and for changes unrelated to the intervention and is thus a suitable tool for evaluating REDD+ effectiveness [10].

While the ideal of an experimental impact evaluation design (or at least robust quasi-experiments) remains highly regarded and their interpretation is relatively straight forward, the challenges to have true control groups become increasingly apparent when studying multicausal and multi-effect dynamics that are the norm in socio-ecological systems. Recently, more sophisticated analytical methods such as Difference-in-Differences (DID) regressions, propensity score matching, and Synthetic Control Method (SCM) [6] have been explored as alternatives to overcome the pitfalls of purposeful selection of imperfect control groups and the challenges to conduct randomized control trials. Additionally, a REDD+ becomes part of a wider range of conservation and development investments, the evaluation of REDD+ interventions is further challenged since funding cannot be easily differentiated from a variety of forest conservation initiatives; consequently, what is contrasted are areas with and without any form of such investments [8,10].

In spite of a vast number of publications dedicated to REDD+, relatively few have evaluated the effectiveness of these programs and even fewer have addressed their effect in reducing forest cover loss. Moreover, since these studies use different evaluation methods, direct comparisons among them have some caveats. In a global study on REDD+ (23 subnational initiatives in six countries), Bos et al. [10] evaluated methods to assess effectiveness in reducing forest cover, including BACI, and found an overall minimal impact at the landscape scale; however, local-scale assessments do show some impacts in lowering deforestation. An evaluation of the Guyana REDD+ national program, applying SCM, revealed an initial effectiveness in reducing the country´s deforestation rate while funding was present (2010–2015), but a reverse trend when discontinued [6]. Subnational REDD+ initiatives in Brazil have been evaluated, mainly through the Amazon Fund projects [13–15]. The absence of a clear strategy for allocation of funds and lack of targeting deforestation hotspots is described as limiting REDD+ effectiveness [13]; plus, projects lack rigorous measurement and evaluation mechanisms to assess performance [14]. Nevertheless, a local-scale evaluation of an individual REDD+ project in the Brazilian Amazon, applying the DID method, found that participating farmers reduced their deforestation by half [15].

In 2010, Mexico embarked on its implementation of REDD+ programs [16], with subnational initiatives focused on early action priority regions, including the Yucatan Peninsula [16,17]. A handful of REDD+ evaluations have assessed multi-level governance capacity and its implications in REDD+ priority regions [18]; the design of pro-poor REDD+ interventions and distribution systems for the Yucatan Peninsula [19]; and challenges and requirements for establishing functional REDD+ monitoring, reporting, and verification systems (MRV) in Mexico [20,21]. Nevertheless, after 10 years since its outset, here, we present the first assessment of REDD+ effectiveness in reducing forest cover loss in the Yucatan Peninsula, a region with the largest tropical forest area in Mexico.

### 1.2. History of REDD+ in Mexico and the Yucatan Peninsula

Mexico's extensive forest cover is important to global climate change mitigation. From 2009 to 2014, Mexico received the third largest portion (11%) of committed funding for REDD+ implementation, joining key tropical forest countries such as Brazil (42%) and Indonesia (33%) [22]. The Yucatan Peninsula in southeast Mexico was assigned as a REDD+ early action priority region [17] since it encompasses a large portion of the Selva Maya which stretches down to Belize and northern Guatemala and constitutes the second largest mass of tropical forest in the Neotropics after the Amazon. Moreover, communal ownership of most of Mexico´s forest lands (around 64% [23]), containing exemplary global cases of sustainable community forest management [24], were viewed as favorable circumstances for the implementation of REDD+ [25,26]. Yet, after a decade of REDD+ in Mexico pursuing the national objective of zero net emissions from deforestation by 2020 [27,28], trends in deforestation and degradation have persisted and even spiked across the country, including on the Yucatan Peninsula [29].

In 2010, the REDD+ strategy in Mexico laid out a series of climate change mitigation and adaptation actions that would be implemented in three phases: (1) preparation and policy design, (2) establishment of finance schemes and pilot projects, and (3) establishment of results-based carbon credit systems for implementing low-carbon land use practices. A landscape approach was taken to pursue low carbon sustainable rural development, focusing on forestry activities [16,17]. Communities were recognized as forest owners who should directly receive benefits, and strategic actions would not threaten land rights and sustainable resource use. Meanwhile, the federal government would act as promotor, regulator, coordinator, mediator, and executioner of REDD+ actions [28].

Subnational activities were initiated in 2010 on the Yucatan Peninsula via three major national and international institutions: the National Forestry Commission (CONAFOR), Alianza México REDD+ (AMREDD), and the United Nations Development Program (UNDP) [30]. AMREDD was formed in 2012, bringing other players on board besides CONAFOR, for example, The Nature Conservancy (TNC), Rainforest Alliance (RA), Woods Hole Research Center, and many national and local nongovernment organizations (NGOs). The REDD+ alliance was fundamental to achieving the objective of institutional strengthening and capacity development of government institutions, NGOs, and communities in priority forested landscapes. Moreover, UNDP began implementing sustainable forest production projects with CONAFOR (e.g., Biodiversidad en Bosques de Producción y Mercados Certificados) and small local-based grants for sustainable rural development practices (GEF-SGP) [30]. A Mexico–Norway Collaboration project (Proyecto Fortalecimiento REDD+ y Cooperación Sur-Sur) was also formed and dedicated mostly to developing the national monitoring, reporting, and verification (MRV) component [31]. REDD+ interventions on the Yucatan Peninsula have mainly centered on strengthening state and municipal jurisdictions for integrating REDD+ policies and implementing low carbon rural development strategies in the region. However, initial activities have also included developing a subnational MRV system, establishing learning communities, and promoting and piloting low carbon land uses in the field (e.g., sustainable forestry, silvopastoral systems, and low-input agricultural intensification and diversification). An approach of working at the local, state, and national levels was adopted by NGOs and CONAFOR for REDD+ implementation [31]. A total of 25 major initiatives (projects) are reported to have been implemented in Mexico with REDD+ -related funding, 10 of which are still ongoing while the rest have concluded [31]. AMREDD reported interventions that covered 133,578 ha of forest regions in Mexico where planning and management mechanisms were implemented, in addition to 13,019 ha with soil and forest conservation projects and 6307 ha with direct investment in sustainable land use practices, benefiting 86 communities [17].

### 1.3. Challenges to REDD+ Implementation and Evaluation

Pursuing REDD+ goals of reducing carbon emissions from forest cover impacts, enhancing sustainable forest management and conservation, and improving rural livelihoods while ensuring social and cultural safeguards and avoiding spill-over effects is a difficult task in complex socioecological landscapes such as Mexico´s Selva Maya. Deforestation and degradation processes in the region have

not been homogenous, and its causes are varied and complex. In our study area covering the Mexican states of Campeche, Quintana Roo, and Yucatan (Figure 1), contrast in forest cover loss have been demonstrated by land change studies [32–34], describing a dynamic forested landscape affected by multiple drivers. Degradation studies have been very few but suggest that it is a much greater factor in forest cover change compared to deforestation [35]. With respect to regional differences in deforestation, the Calakmul Biosphere Reserve region in southern Campeche state is the most researched and is characterized by low rates aided by the protected area presence [36]. In the central region of Quintana Roo, forest cover maintenance is facilitated by a wide-spread occurrence of community forests managed by local communities for economic and subsistence needs [34,37] and by protected areas (e.g., Calakmul, Sian Kaan, and Balam Kaax). At the other extreme, high deforestation regions are identified in central and southeastern Campeche and southern Yucatan and Quintana Roo in addition to northern coastal regions of the Yucatan Peninsula [38,39]. Rural areas have been impacted by expanding land uses for cattle raising and commercial mechanized agriculture [35], while infrastructure growth in urban and tourist centers drive deforestation, particularly in coastal areas [39]. Trying to sort out and distinguish what are the impacts of REDD+ interventions in reducing forest cover loss from a diverse range of regional and local forest cover change trends and drivers is thus truly challenging.

Other challenges have surfaced during REDD+ implementation in México and the Yucatan Peninsula. At the local level, participation and benefits from REDD+ pilot projects have been described as benefiting communities with greater forest resources and mostly households with rights to land and forest benefits [26,40], having shortfalls in equity and implementing safeguards for a variety of other "communities" and households that impact forest cover. Skutsch and Turnhout [41] argue that REDD+ national projects may be focusing on the wrong actors (or deforestation drivers), with greater attention given to communities with small-scale slash and burn agriculture and traditional forest practices while largely ignoring commercial and industrial agriculture as major deforestation drivers. Trying to implement a multi-level governance approach to pursue REDD+ at the local and subnational levels has also been difficult, since municipal governments have very limited influence on land-use decision-making and regulation, and state and federal governments maintain a lopsided share of authority and control over land use [42]. Unfortunately, at the federal political and economic levels of REDD+ implementation, there is a lack of government commitment towards transitioning to low carbon development in Mexico [27]. Further, although the 2016 tri-state (Yucatan, Campeche, and Quintana Roo) accord for sustainable development (ASPY) laid out a promising jurisdictional and institutional environment for future REDD+ implementation at the subnational level, the accord was legally challenged in 2018 by a local indigenous organization claiming that Mayan communities neither had been previously informed nor had given their consent.

A variety of research have evaluated the REDD+ mechanism in Mexico and the Yucatan Peninsula region at the national and subnational levels. However, the majority of evaluations have focused on how multi-level governance has functioned towards achieving REDD+ goals [18,20,43], reporting disjointed and limited subnational level jurisdiction which reduces the efficacy of REDD+ implementation in Mexico. Other research describes how well land use and cultural diversity, social equity, and pro-poor strategies are integrated into its initial actions and the implications on its success [19,44]. After a decade of preparing and initiating REDD+ in Mexico, research has yet to evaluate if any impacts on reduced forest cover can be noticed in intervened areas at the subnational and local levels or how current jurisdictional and implementation challenges affect REDD+ effectiveness in reducing deforestation. At this junction, it is important to determine the forest cover outcomes of REDD+ and the elements of success and failure to inform ongoing initiatives and future planning and strategies [10].

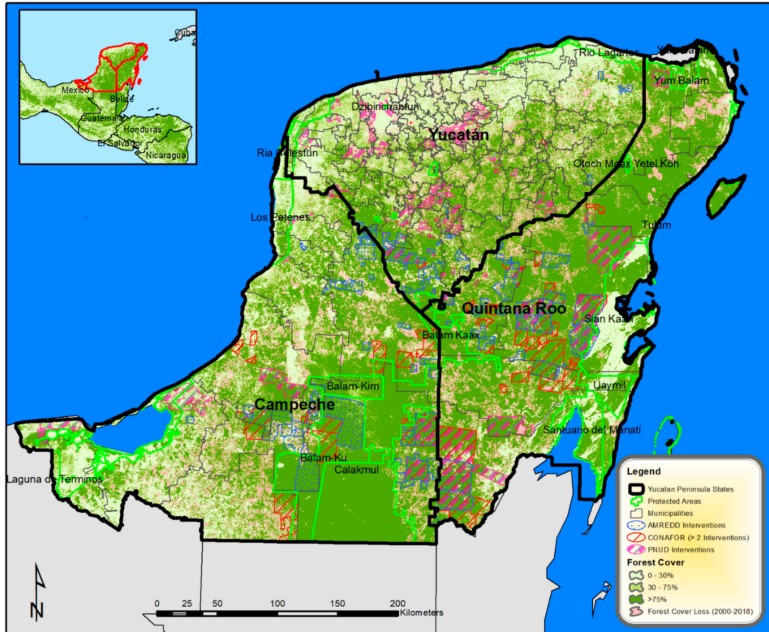

**Figure 1.** Yucatan Peninsula study area showing 2010 forest cover [45], deforestation (2000–2018) [38], municipalities, and REDD+ (Reducing Emissions from Deforestation and Degradation)-intervened community territories (ejidos).

### 1.4. Resarch Objetives and Methodological Approach

The goal of this study was to assess the effectiveness of REDD+ in reducing forest cover loss impacts in the Selva Maya landscape of southeastern Mexico after ten years of interventions. We applied quasi-experimental methods and the BACI approach to evaluate the impacts on forest cover loss that may be attributed solely to REDD+ interventions in municipalities and rural forest communities (ejidos) on the Yucatan Peninsula (Figure 1). Thus, we consider and evaluate REDD+ effectiveness at the regional (meso) and community (micro) scales as suggested by Bos et al. [10]. Considering the diversity of forest cover loss trends among municipalities and ejidos, the approach and methods applied allow for statistical comparisons between intervened and non-intervened municipalities and communities, considering similar counterfactual (control) cases. Our results bolster previous findings and arguments about the challenges and weak points of implementing REDD+ with cattle and agricultural producers. However, findings also show some successes in key municipalities and communities that are worth noting. Finally, our study reveals important information that provide insight and guidance to the continued implementation of REDD+.

## 2. Materials and Methods

### 2.1. Study Area

The Yucatan Peninsula in Mexico (Figure 1) spans 142,310 km$^2$ of forested landscape; tropical forests cover 60% of the territory [46]. Forest and coastal ecosystems contain more than 4000 species: over 2000 plant, 300 trees, 400 birds, and 100 mammals [47,48]. Close to two million hectares are under federal protected status on the Yucatan Peninsula, and two large UNESCO Biosphere Reserves, Calakmul and Sian Kaan (over one million hectares combined) [49], are joined by a large swath of inhabited and managed tropical forests which form part of the extensive Mesoamerican Biological Corridor [50]. Other major protected areas include Balam Ku, Balam Kaax, and Balam Ki in central and southern inland moist forest areas and Los Petenes, Rio Lagartos, and Yum Balam in western and northern coastal dry forest areas.

Of the 7.3 million hectares of forest cover, over half (60%) is considered tropical moist forest, which extends over the southern and eastern portions of the Yucatan Peninsula. The remaining forest cover, mostly dry tropical forest, is distributed in the western and northern portions. These forests are subject to frequent disturbances from natural and human causes, creating mosaics of successional secondary and disturbed forest that make up close to 80% of forest cover; only 20% is regarded as old growth or intact forest [46]. The cultural and economic significance of the Selva Maya on the Yucatan Peninsula goes hand in hand, based on a rich history that revolves around the use and commercialization of forest products [37]. As its name implies, the region has been inhabited by Mayans for millennia, leaving deep-seated traditions and practices in natural resource management, such as milpa slash and burn agriculture and silviculture for a variety of timber and non-timber species [51].

After the Mexican Revolution (1917–1921), the agrarian reform set out to establish territories with common property (ejidos) for production activities by rural communities. Ejidos presently dominate the landscape of the Yucatan Peninsula. Early ejidos allotted in the 1930s and 1940s were large and forestry-oriented landholdings, particularly intended for chicle (chewing gum resin) extraction, a major economic export at the time, and for timber. After the 1960s, ejidos were smaller and established for agricultural production rather than forestry [37]. Agricultural development and colonization policies during the 1970s up to the 1990s resulted in large-scale deforestation, which has fluctuated but persisted on the Yucatan Peninsula [29]. The region has become a global tourist hub, with attractions such as the Mexican Caribbean in Quintana Roo, Mayan archeological sites and culture, and historical colonial cities, such as Campeche City and Merida, Yucatán. Recent expansion of commercial mechanized agriculture has spread in the states of Campeche and Quintana Roo, in part due to growing Mennonite settlements immigrating since the 1990s [38]. Cattle ranching is a prominent land use in southeast Mexico and a major deforestation and degradation driver [35,38]. Many ejido communities still practice traditional slash and burn agriculture, mostly for subsistence, and rely on community forests for important economic activities such as bee-keeping and timber management [35].

## 2.2. Data Sources and Preparation

Annual forest cover loss data [38] from 2000 to 2018 was downloaded from the University of Maryland's Global Forest Change (GFC) 2000–2018 website [52]. On the Yucatan Peninsula, this data also includes temporary forest cover losses (52 to 64%) that can occur from natural disturbances, such as fire, to human-induced slash and burn agriculture [32,35]. Consequently, we recognize that the GFC data does not solely reflect deforestation processes but can also include forest degradation and nonpermanent tree cover loss [35]. Nevertheless, GFC data, which is based on Landsat imagery (30 m resolution), is widely used as an effective indicator of forest cover loss in a variety of land change and policy research [10,53]. Comparisons in this study are thus made of tree cover loss between REDD+ -intervened and non-intervened municipalities and communities and do not specify deforestation. We assume that any tendencies in the errors of the GFC data used in our study are the same for the entire Yucatan Peninsula, allowing for unbiased comparisons of municipalities and ejidos at the regional and local scales, respectively. For our quasi-experimental analyses, total forest cover loss in municipalities and ejidos were calculated for the periods 2000 to 2004, 2005 to 2009, 2010 to 2014, and 2015 to 2018. Subsequently, indicators representing yearly rates of loss based on percent unit area of analysis (municipality or ejido) were derived for each period and used as dependent variables for BACI analyses. The municipal and community forest loss indicator rates used in this research do not represent actual deforestation rates and should not be used for reporting or making regional, national, or global comparisons.

The other key dataset produced for this research involves the identification and georeferencing of REDD+ interventions. The REDD+ -intervened municipalities and communities (ejidos) were based on the presence of REDD+ -related and -funded projects. Sources used to identify and locate REDD+ projects varied, starting with data provided by CONAFOR on financial support given to

ejido communities for community forest management and conservation programs from 2010 to 2018 (e.g., management plan development, environmental impact statements, training, reforestation, and enrichment planting). Although forest management has been a land use practice supported by CONAFOR before the REDD+ initiative in Mexico, after 2010, forest management and conservation programs were integrated into the REDD+ mechanism and were largely funded by the World Bank project [30,31]. Secondly, information was obtained for projects funded by the UNDP GEF-SGP from 2010 to 2018 (e.g., climate change and biodiversity) [54]. Thirdly, we included projects that were funded through AMREDD aimed at installing REDD+ on the Yucatan Peninsula and were led by TNC in collaboration with other major international and national NGOs. Georeferenced data of AMREDD community-based interventions on the Yucatan Peninsula were provided by TNC and were confirmed by other sources [17,30,31,55,56].

The criteria used to select the REDD+ interventions were projects (1) implemented after 2010, (2) located exclusively on the Yucatan Peninsula at the municipal or local level, (3) implemented in and related to terrestrial forest ecoregions (i.e., excluding coastal or fisheries projects), and (4) dealing with sustainable forestry, agricultural and cattle production, rural community development (e.g., ecotourism, community enterprises), biodiversity conservation, or local governance subjects. For the selected REDD+ communities, at least three CONAFOR programs and/or one GEF-SGP or AMREDD project had to be present after 2010, and for selected REDD+ municipalities, at least four projects had to be present with the exception of Yucatan with much smaller municipalities, where one project was sufficient. In this manner, we exclude large municipalities with minimal REDD+ activity or weak CONAFOR presence. Our final sample of treated or REDD+ -intervened municipalities was 39 and included (1) 7 out of 11 in Campeche, (2) 5 out 10 in Quintana Roo, and (3) 27 out of 106 in Yucatan. For communities (ejidos), a single project was sufficient to be considered "treated". The total sample was 1368 ejidos, of which 140 were REDD+ -intervened communities: (1) 36 out of 374 in Campeche, (2) 48 out of 281 in Quintana Roo, and (3) 56 out of 713 in Yucatan.

Drivers of deforestation involve multiple environmental, economic, cultural, or policy conditions that act from the global to local scales [57]. For that reason, identifying and comparing counterfactuals is challenging, requiring the integration of a variety of socioeconomic and spatial covariates that may influence forest cover loss at both the regional and local scales. In this study, we integrate a set of variables representing demographic, socioeconomic, tenure, and institutional features for both municipalities and communities. Table 1 summarizes the covariates used in the quasi-experimental analyses to reduce the effect of possible confounding factors. Demographic variables such as total population, population density, and migrant and indigenous populations are included. Socioeconomic statistics on poverty conditions, immigration, remittances, and employment and data on agriculture and cattle production are also integrated in our analyses. Institutional characteristics relate to land tenure characteristics (e.g., private property in ejidos and private agriculture and cattle land). In addition, forestry activities and production before 2010 are also associated with institutional conditions due to the strong support from government programs (CONAFOR) to keep community forest management in place. Furthermore, environmental and spatial variables including elevation, fire density, distance to roads, and urban areas are integrated for the local ejido-scale analyses. For the final covariate selection used in our analyses (Table 1), a Pearson´s correlation test was previously applied to more than 50 potential covariates to detect and remove highly correlated variables above 0.75. The final set of covariates selected are all frequently used in deforestation driver studies and are common examples of direct and underlying factors affecting land use change and deforestation in the tropics [58].

**Table 1.** Covariates used in quasi-experimental analyses Difference-in-Differences (DID), Propensity Score Matching (PSM), and Synthetic Control Method (SCM)) of REDD+ effectiveness in reducing forest cover loss at meso (municipal) and micro (ejido community) scales on the Yucatan Peninsula.

| Variable | Category | Scale | Description |
|---|---|---|---|
| Population | Socioeconomic | Municipality and Ejido | 2010 population [59] |
| PopDens | Socioeconomic | Municipality and Ejido | Population density per municipality or ejido area [59] |
| BornOutState | Socioeconomic | Municipality and Ejido | Percent population born outside the state [59] |
| OutState05 | Socioeconomic | Municipality | Percent population of out of state residents arriving after 2005 [59] |
| IndHH | Socioeconomic | Municipality and Ejido | Percent indigenous speaking households [59] |
| Unemployed | Socioeconomic | Municipality and Ejido | Percent population without an occupation (non-agricultural) [59] |
| Employed | Socioeconomic | Municipality and Ejido | Percent economically active population with formal employment [59] |
| UrbPopClass | Socioeconomic | Municipality | Index of urbanization and population by municipality [60] |
| Resident05 | Socioeconomic | Municipality and Ejido | Percent of population residing before 2005 [59] |
| MargIndex | Socioeconomic | Municipality and Ejido | Index of socioeconomic marginalization or poverty conditions [61] |
| MigraIndex | Socioeconomic | Municipality | Migration index by municipality based on demography and remittance data [62] |
| TimbHa | Institutional/forestry | Municipality and Ejido | Timber management ANNUAL CUTTING areas (ha) per ejido [63] |
| VolTimb | Institutional/forestry | Municipality and Ejido | Annual authorized timber harvest volume per ejido [63] |
| PrivParc | Institutional/tenure | Municipality and Ejido | Area (ha) of ejido land parceled with private ownership [64] |
| PubProp | Institutional/tenure | Municipality | Area (ha) of public land in municipality [65] |
| AgrPer | Socioeconomic | Municipality DID, PSM | Percent land under agricultural production in municipality [65] |
| CatpastPer | Socioeconomic | Municipality | Percent land under pasture for cattle raising [65] |
| AgrHa | Socioeconomic | Municipality and Ejido | Area (ha) of land under crop or pasture [65] |
| AgrHaPerc | Socioeconomic | Municipality and Ejido | Percent area under crop or pastureland [65] |
| AgrPrivProp | Socioeconomic | Municipality | Percent area of private agricultural land in municipality [65] |
| CatPastPriv | Socioeconomic | Municipality | Percent private pastureland for cattle in municipality [65] |
| ValAgrTot | Socioeconomic | Municipality | Total value of agricultural/cattle production in municipality [65] |
| DistRds | Socioeconomic | Ejido | Distance to roads [66] |
| DistUrb | Socioeconomic | Ejido | Distance to urban areas [66] |
| AvgElev | Environmental | Ejido | Elevation in meters [67] |
| FireDens | Environmental | Municipality and Ejido | Density of fires recorded from 2000 to 2018 [68] |

*2.3. Before-and-After-Control Intervention*

This study adopts a Before-After-Control-Intervention (BACI) or Difference-in-Differences (DID) approach to evaluate REDD+ effectiveness in halting forest cover loss on the Yucatan Peninsula. The BACI method is included as an essential research tool for REDD+ subnational initiatives [11] and was evaluated by Bos et al. [10], testing REDD+ effectiveness in reducing carbon emissions in six tropical countries. For this research, we also compare tree cover loss between two periods, before and after REDD+ interventions, and evaluate differences between intervened and control units, at the regional and community (ejido) scales. Finding suitable control units is the challenging part of the BACI method [10]. We apply counterfactual thinking and quasi-experimental methods as suggested by Ferraro [8] for studies that evaluate environmental policy; while having its limitations, the use of quasi-experimental designs provide a practical means to overcome the difficulties in identifying similar controls for comparison. Three different DID methods are used to test the effectiveness of REDD+ subnational initiatives at the municipal and community (ejido) scales in our study area: (1) regression modelling, integrating a variable of treatment (intervention) interacting with time (before and after); (2) propensity score matching, used to identify counterfactuals for statistical comparison; and (3) the synthetic control method, which produces its own "synthetic" counterfactual in order to evaluate specific individual cases (municipality or ejido).

2.3.1. DID Regression

DID regression is a quasi-experimental design often used to assess the causal relationships of policy interventions on expected outcomes when randomized control trials are not feasible or integrated into the implementation programs [69]. The DID regression model compares two groups (control and treated) that are assumed to experience the same deforestation pressures, shocks, and trends before and after the intervention period. The DID regression model statistically evaluates the outcomes of the difference between forest loss after and before the intervention in the control and treated groups [70]. If there is an association between the time of REDD+ intervention and forest loss outcome, then the interaction term should be significant [69,70]. Under the assumption of parallel trends, the effects of other factors on forest loss are presumably stripped away. The regression model also allows for the introduction of important covariates to control and test their effects on forest loss, as in other multi-variate regression models [69,71]. The DID regression method has been applied and described in a variety of economic, social, and health science studies [71–74]. We used the Tidyverse package in RStudio 1.2.5 for data and statistical analysis. The lm() function was used, integrating the DID interaction between time (before and after) and REDD+ intervention, called DID = data$time × data$redd. The covariates included in the DID regression model are indicated in Table 1.

2.3.2. Propensity Score Matching

Propensity Score Matching (PSM) is a statistical technique used in quasi-experimental designs in which treated cases are matched with control cases based on their propensity scores [75]. The propensity score is a measure of a sample unit´s probability of being assigned the treatment based on a set of observed covariates. The matching produces a set of control units that can be used as counterfactuals for BACI comparison and assessment of treatment effectiveness, ensuring that the treatment is the cause for the difference and strengthening causal arguments [76]. We used PSM to identify municipal and community (ejido) control units for comparison with REDD+ -intervened municipalities and ejidos. We used the MatchIt package in RStudio 1.2.5 for PSM using the matchit() function. The covariates included in the model are specified in Table 1. The intervened and non-intervened units were then compared, and the significance of difference tested using nonparametric Mann–Whitney U test for two sample comparison due to the nonnormality of dependent variable. XLStat2019 was used for Mann–Whitney U test.

### 2.3.3. Synthetic Control Method

The assumption of a parallel trend between treated and non-treated units for DID regression is hard to verify as well as the accuracy of the control group selected by PSM to represent counterfactuals [77]. For this reason, we also apply the Synthetic Control Method (SCM) to test the effect of REDD+ interventions in reducing forest cover loss in a single municipality or ejido. For the analysis of REDD+ community cases, the SCM was conducted separately by state (i.e., Campeche, Quintana Roo, and Yucatan). SCM creates a "synthetic" counterfactual from a group of similar untreated units by selecting a weighted average of the outcome variable [77]. It is an empirical and data-driven method used to evaluate a treatment in an individual case, applying bootstrapping and placebo tests [78]. SCM is an already proven method, applied by Roopsind et al. [6] to assess the impact of REDD+ interventions in reducing deforestation in the country of Suriname. In Brazil, SCM was used to evaluate deforestation policies and programs in the municipality of Paragominas [78]. Both studies claim improvements in using SCM over other quasi-experimental methods for analyzing policy to reduce deforestation. For example, SCM is suitable when evaluating large jurisdictional areas, akin to municipalities on the Yucatan Peninsula. Besides matching covariates, the average outcome of interest before the intervention is matched by applying linear combinations to control for unobserved factors of forest cover loss over time [6,78]. We used the Synth package in RStudio 1.2.5 for SCM using the synth() function. The covariates included in the model are specified in Table 1.

## 3. Results

### 3.1. Municipal (Meso) Scale Analyses

The DID regression model (Table 2) employed to test REDD+ effectiveness at the municipal scale did not show a significant effect of REDD+ project interventions in reducing forest cover loss after 2010 (DID = Time × REDD interaction term, $p = 0.219$). Before REDD+ implementation (2000–2009), mean annual rate of forest cover loss was 0.82 (SD = 0.45) in non-intervened municipalities and 0.73 (SD = 0.40) in REDD+ intervened municipalities. After REDD+ implementation (2010–2018), mean annual forest cover loss decreased to 0.48 (SD = 0.37) in non-intervened and 0.46 (SD = 0.27) in REDD+ intervened municipalities. Change in forest cover rates before and after REDD+ interventions are also similar in non-intervened (mean = −0.34, SD = 0.39) and intervened (mean = −0.27, SD = 0.31) municipalities. The overall DID model of forest cover loss in municipalities was significant ($F = 10.64$, df = 228, $p < 0.0001$) and explained half of the variance in forest cover loss ($R^2 = 0.54$). Significant covariates associated with forest cover loss included fire density (FireDens, $p < 0.0001$), population density (PopDens, $p < 0.0008$), percent agricultural area (AgrPer, $p = 0.003$), out-of-state population residing after 2005 (OutState05, $p < 0.03$), and population-urbanization index (UrbPopClass, $p = 0.04$).

Difference-in-differences between REDD+ and non-REDD+ municipalities was also evaluated using a set of 42 matched control municipalities selected by PSM (Figure 2). The matched control units were compared to 42 treated units (REDD+ intervened municipalities) applying Mann–Whitney U test. Change in forest cover rates in non-intervened municipalities (mean = −0.28, SD = 0.36) and intervened (mean = −0.27, SD = 0.32, Figure 3) municipalities before and after REDD+ funded projects were very similar and not significantly different ($U = 824.5$, $p = 0.61$). Both DID and PSM results show an overall tendency that REDD+ municipalities have similar forest cover loss trends as non-intervened municipalities, indicating no effect of project interventions in affecting forest cover loss rates at the regional level.

**Table 2.** Difference-in-Difference (DID) regression model results for municipal-scae analysis of REDD+ effectiveness in reducing forest cover loss on the Yucatan Peninsula (* $p < 0.05$).

| Variable | Coefficients | Std. Error | t Value | Pr (>*t*) |
|---|---|---|---|---|
| (intercept) | $-3.385 \times 10^0$ | $2.157 \times 10^0$ | $-1.569$ | 0.118030 |
| Time | $-3.365 \times 10^0$ | $4.671 \times 10^{-2}$ | $-7.204$ | $8.43 \times 10^{-12}$ |
| REDD | $-1.429 \times 10^{-1}$ | $6.314 \times 10^{-2}$ | $-2.263$ | 0.024573 * |
| DID | $1.008 \times 10^{-1}$ | $8.185 \times 10^{-2}$ | $1.232$ | 0.219173 |
| Population | $-2.953 \times 10^{-7}$ | $3.491 \times 10^{-7}$ | $-0.846$ | 0.398516 |
| MargIndex | $9.022 \times 10^{-2}$ | $8.496 \times 10^{-2}$ | $1.062$ | 0.289380 |
| Bornoutstate | $5.011 \times 10^{-3}$ | $3.921 \times 10^{-3}$ | $1.278$ | 0.202515 |
| Outstate05 | $9.975 \times 10^{-2}$ | $4.461 \times 10^{-2}$ | $2.236$ | 0.026323 * |
| IndHH | $-4.871 \times 10^{-4}$ | $1.307 \times 10^{-3}$ | $-0.373$ | 0.709739 |
| Unemployed | $-2.948 \times 10^{-2}$ | $1.965 \times 10^{-2}$ | $-1.500$ | 0.135038 |
| UrbPopClass | $5.027 \times 10^{-2}$ | $2.432 \times 10^{-2}$ | $2.067$ | 0.039834 * |
| Employed | $-4.084 \times 10^{-2}$ | $2.137 \times 10^{-2}$ | $-1.911$ | 0.057226 |
| Resident05 | $6.878 \times 10^{-2}$ | $3.611 \times 10^{-2}$ | $1.905$ | 0.058033 |
| PodDens | $1.090 \times 10^{-1}$ | $3.200 \times 10^{-2}$ | $3.408$ | 0.000775 * |
| MigraIndex | $8.218 \times 10^{-2}$ | $5.005 \times 10^{-2}$ | $1.642$ | 0.101983 |
| PrivParc | $-1.431 \times 10^{-5}$ | $2.188 \times 10^{-3}$ | $-0.007$ | 0.994788 |
| PubProb | $-2.179 \times 10^{-3}$ | $4.269 \times 10^{-3}$ | $-0.511$ | 0.610193 |
| AgrPer | $5.267 \times 10^{-3}$ | $1.761 \times 10^{-3}$ | $2.992$ | 0.003078 * |
| CatpastPer | $7.839 \times 10^{-4}$ | $2.093 \times 10^{-3}$ | $0.375$ | 0.708319 |
| AgrHa | $6.803 \times 10^{-6}$ | $2.496 \times 10^{-5}$ | $0.273$ | 0.785477 |
| AgrPrivProp | $-2.55 \times 10^{-3}$ | $4.334 \times 10^{-3}$ | $-0.590$ | 0.555988 |
| CatpastPriv | $-4.152 \times 10^{-3}$ | $3.677 \times 10^{-3}$ | $-1.129$ | 0.260042 |
| VolTimb | $3.010 \times 10^{-10}$ | $3.494 \times 10^{-10}$ | $0.861$ | 0.389917 |
| TimbHa | $-1.999 \times 10^{-8}$ | $2.390 \times 10^{-8}$ | $-0.836$ | 0.403809 |
| ValAgrTot | $1.244 \times 10^{-10}$ | $2.767 \times 10^{-10}$ | $0.450$ | 0.653419 |
| FireDens | $7.248 \times 10^{-5}$ | $1.008 \times 10^{-5}$ | $7.190$ | $9.20 \times 10^{-12}$ |

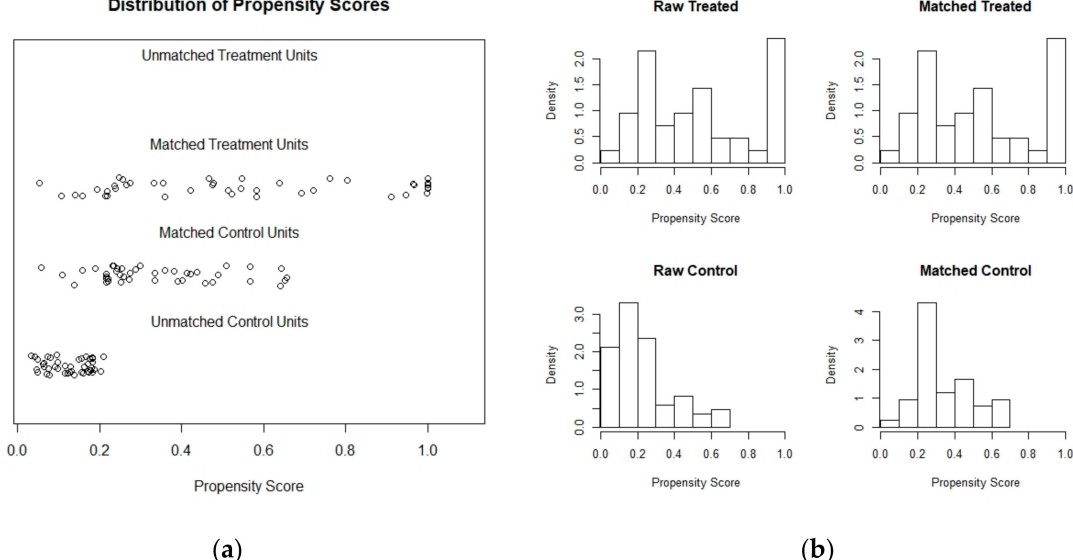

(**a**)　　　　　　　　　　　　　　　　　　　　　　　(**b**)

**Figure 2.** PSM results showing matched treated (REDD+) and unmatched control municipalities or units (**a**) and distributions of raw and matched treated and control units or municipalities (**b**).

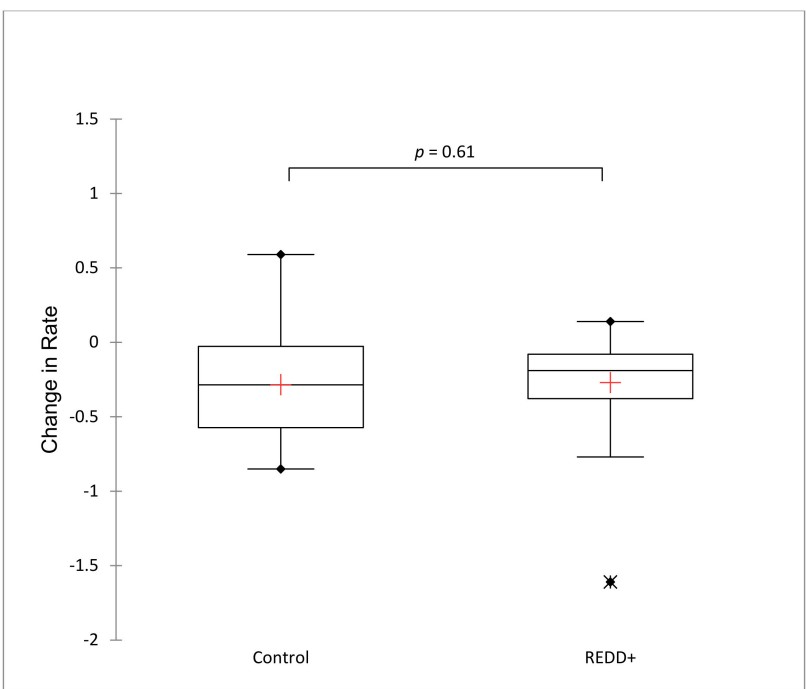

**Figure 3.** Boxplots and Mann–Whitney U two-sample comparison of change in the annual rate of forest cover loss after REDD+ implementation (2010–2018) in matched control municipalities derived from propensity score matching (0) and treated or REDD+ intervened municipalities (1).

The SCM analyses provided an additional nonparametric test of REDD+ effectiveness in single treated units (municipalities), providing more detailed results that allow for the detection of patterns and trends associated with decreasing forest cover loss, or no reduction and increasing rates of forest cover loss. Table 3 summarizes SCM outputs for all REDD+ municipalities (mapped in Figure 4) indicating a range of mixed effectiveness in reducing forest cover loss. REDD+ intervention showed no effect in reducing forest cover loss (i.e., no change or increase in deforestation rate) after 2010 in half (20) of the 39 intervened municipalities. Of these municipalities, 14 are in Yucatan state, which contains 83% of all municipalities and are much smaller than in Campeche or Quintana Roo. Notable municipalities in the state of Yucatan showing an absence of REDD+ impacts include Celestun, Chacsinkin, Hunucma, Izamal, Merida, Oxkutzcab, Tekax, and Tizimin. Tizimin is a highly deforested municipality with cattle production land uses, and Merida seats the capital which is a major growing urban center of the Yucatan Peninsula. Municipalities of Tekax, Chacsinkin, and Oxkutzcab in the south are in a deforestation hotspot with expanding commercial agriculture identified as a REDD+ priority region, the Jibio Puuc Biocultural Reserve. Four municipalities in Campeche state (Calakmul, Campeche, Carmen, and Hopelchen) and two in Quintana Roo (Othon P. Blanco and Jose María Morelos) had no effect from REDD+ interventions in reducing rates of forest cover loss. Carmen is a municipality with widespread deforestation for cattle production, while Hopelchen and Campeche are cases of expanding commercial mechanized agriculture. A net forest cover loss and REDD+ ineffectiveness in Calakmul is concerning due to protected area presence and greater investment in REDD+ interventions in the municipality. In Quintana Roo, Othon P. Blanco has been deforested for cattle and sugarcane production while commercial agriculture has expanded in José María Morelos, deterring REDD+ effectiveness. A pattern is observed of municipalities that lack reduced forest cover loss and REDD+ effectiveness located in the southern Yucatan Peninsula region and border zone with Belize and Guatemala.

**Table 3.** Synthetic Control Method (SCM) results of treated unit effects for each REDD+ municipality (negative sign indicates reduction in deforestation rate) (CONAFOR = National Forestry Commission, UNDP = United Nations Development Program, AMREDD = Mexico REDD+ Alliance).

| Municipio | State | Effect | REDD+ Interventions |
|---|---|---|---|
| Calakmul | Campeche | 0 | CONAFOR, UNDP, AMREDD |
| Calkini | Campeche | −0.1 | UNDP |
| Campeche | Campeche | 0.2 | UNDP |
| Carmen | Campeche | 0.4 | UNDP, AMREDD |
| Champotón | Campeche | −0.2 | CONAFOR, UNDP, AMREDD |
| Escárcega | Campeche | −0.4 | CONAFOR, AMREDD |
| Hopelchén | Campeche | 0 | CONAFOR, AMREDD |
| Felipe C. Puerto | Quintana Roo | −0.1 | CONAFOR, UNDP, AMREDD |
| José María Morelos | Quintana Roo | 0.1 | CONAFOR, UNDP, AMREDD |
| Lázaro Cárdenas | Quintana Roo | −0.25 | CONAFOR, UNDP |
| Othón P. Blanco | Quintana Roo | 0.2 | CONAFOR, UNDP, AMREDD |
| Tulum | Quintana Roo | −0.2 | UNDP |
| Cantamayec | Yucatan | −0.05 | UNDP |
| Celestún | Yucatan | 0 | UNDP |
| Chacsinkín | Yucatan | 0.05 | UNDP |
| Maní | Yucatan | 0.4 | UNDP |
| Mayapán | Yucatan | −0.15 | UNDP |
| Mérida | Yucatan | 0.1 | UNDP |
| Oxkutzcab | Yucatan | 0 | UNDP, AMREDD |
| Sinanche | Yucatan | −0.7 | UNDP |
| Sudzal | Yucatan | 0 | UNDP |
| Teabo | Yucatan | −0.2 | UNDP |
| Tekal de Venegas | Yucatan | −0.1 | UNDP |
| Tekax | Yucatan | 0.2 | CONAFOR, UNDP, AMREDD |
| Tetiz | Yucatan | 0 | UNDP |
| Tixcacalcupul | Yucatan | 0.1 | UNDP |
| Tixmehuac | Yucatan | −0.1 | UNDP |
| Tizimin | Yucatan | 0.5 | AMREDD |
| Tzucacab | Yucatan | −0.1 | UNDP, AMREDD |
| Ucú | Yucatan | 0 | UNDP |
| Umán | Yucatan | −0.1 | UNDP |
| Yaxcabá | Yucatan | −0.7 | UNDP, AMREDD |
| Chocholá | Yucatán | −0.05 | UNDP |
| Dzilam de Bravo | Yucatán | −0.05 | UNDP |
| Dzoncauich | Yucatán | 0 | UNDP |
| Halachó | Yucatán | −0.4 | UNDP |
| Hocabá | Yucatán | −0.3 | UNDP |
| Hunucmá | Yucatán | 0 | UNDP |
| Izamal | Yucatán | 0.2 | UNDP |

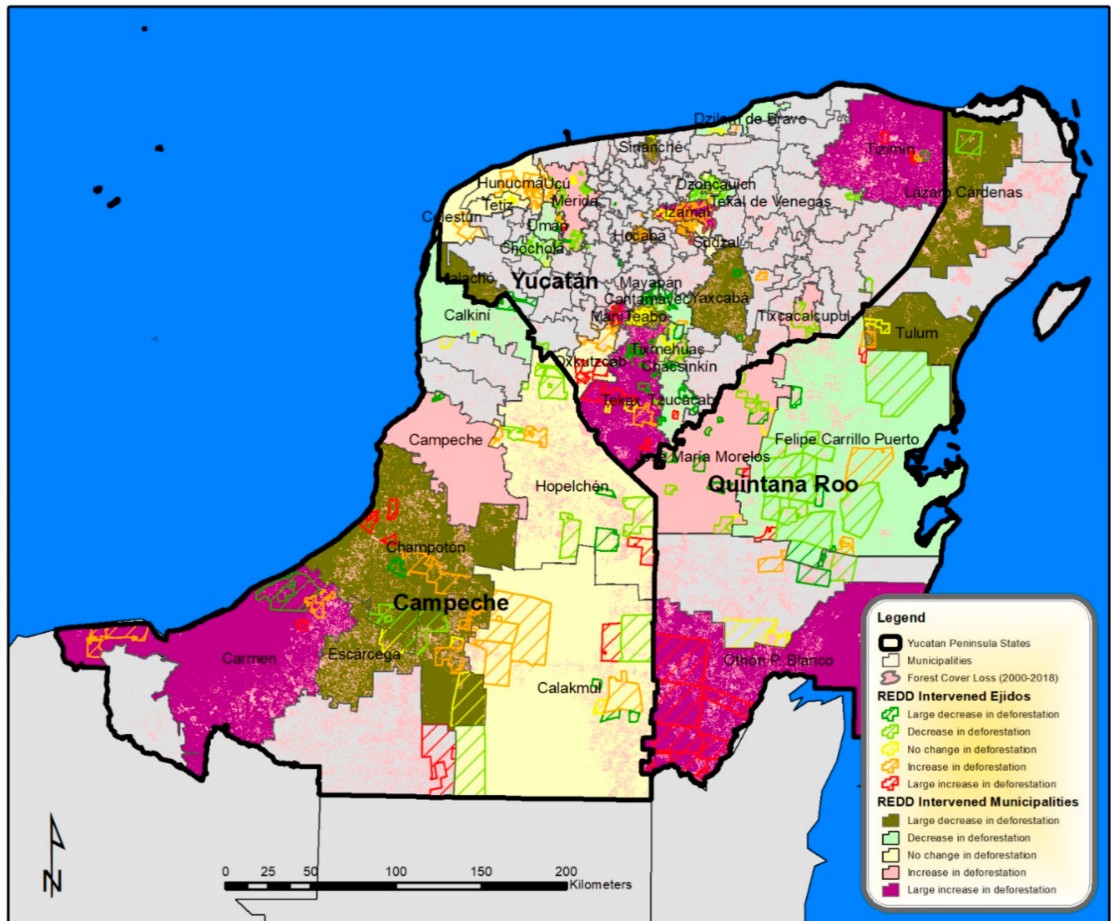

**Figure 4.** REDD+ effectiveness in reducing deforestation in selected intervened municipalities and intervened ejido territories on the Yucatan Peninsula.

On the other hand, potential REDD+ effectiveness was found in the remaining half of REDD+ -intervened municipalities, these included 13 in Yucatan, 3 in Campeche, and 3 in Quintana Roo. In Campeche, municipalities with historic deforestation for agricultural and cattle production, such as Calkini, Escarcega, and Champoton, indicated reduced tree cover loss associated with REDD+ interventions when compared to their synthetic matches. The latter two municipalities have an important role in the state´s forestry sector. In Quintana Roo, municipalities that show REDD+ effectiveness, Felipe Carrillo Puerto, Lazaro Cardenas, and Tulum, are important to maintaining carbon reserves in the landscape. Felipe Carrillo Puerto has low deforestation rates with extensive forest management and subsistence agriculture land uses, while Tulum also has low deforestation and predominant land uses of subsistence agriculture and some forestry. The municipality of Lazaro Cárdenas has had greater deforestation for cattle raising but is also near the major urban and tourism hub of Cancun. Municipalities with REDD+ effectiveness in the state of Yucatan included cases such as Cantamayec, Tekal de Venegas, Tzucacab, Uman, and Yaxcaba. Among REDD+ effective municipalities in Yucatan state were large forested areas in the central and eastern region and surrounding the urban municipality of Merida.

Table 4 indicates that more funders or projects intervening in a municipality do not necessarily result in REDD+ effectiveness. Municipalities receiving the most attention by all three major REDD+ funders (AMREDD, CONAFOR, and UNDP) and with greater number of projects (363) did not show improvements in reducing their rates of deforestation. These include important municipalities for forest conservation: Calakmul and Champoton in Campeche, José María Morelos and Othon P. Blanco in Quintana Roo, and Tekax in Yucatan. Greater REDD+ effectiveness was found among municipalities where AMREDD and CONAFOR or UNDP projects were both present, and the least effectiveness was

found where the smallest number of projects were present. Nevertheless, there does not appear to be any clear or strong positive trend in the level of project effort and REDD+ effectiveness. Major land use cover and productive activities present in municipalities are apparently more associated with REDD+ effectiveness at the meso scale as noted above. Figure 5 shows the average REDD+ effect on forest cover loss rates among municipalities categorized according to their dominant productive land uses. These results show that REDD+ interventions in municipalities where subsistence agriculture and forestry are major land uses tend to be effective in reducing deforestation. Also, interventions in municipalities with cattle, commercial agriculture, and forestry land uses combined performed well. However, in municipalities where commercial agriculture dominates (with the presence of mechanized farming), deforestation was not reduced. Even when forestry production is present, these municipalities failed to show a reduction in the rates of forest cover loss from REDD+ interventions. In municipalities where cattle production dominates, there was on average a small reduction in the rate of forest cover loss, representing municipalities such as Lazaro Cardenas in Quintana Roo and Dzilam de Bravo in Yucatan. Yet, half of these cattle ranching municipalities did not show REDD+ effectiveness in reducing forest cover loss.

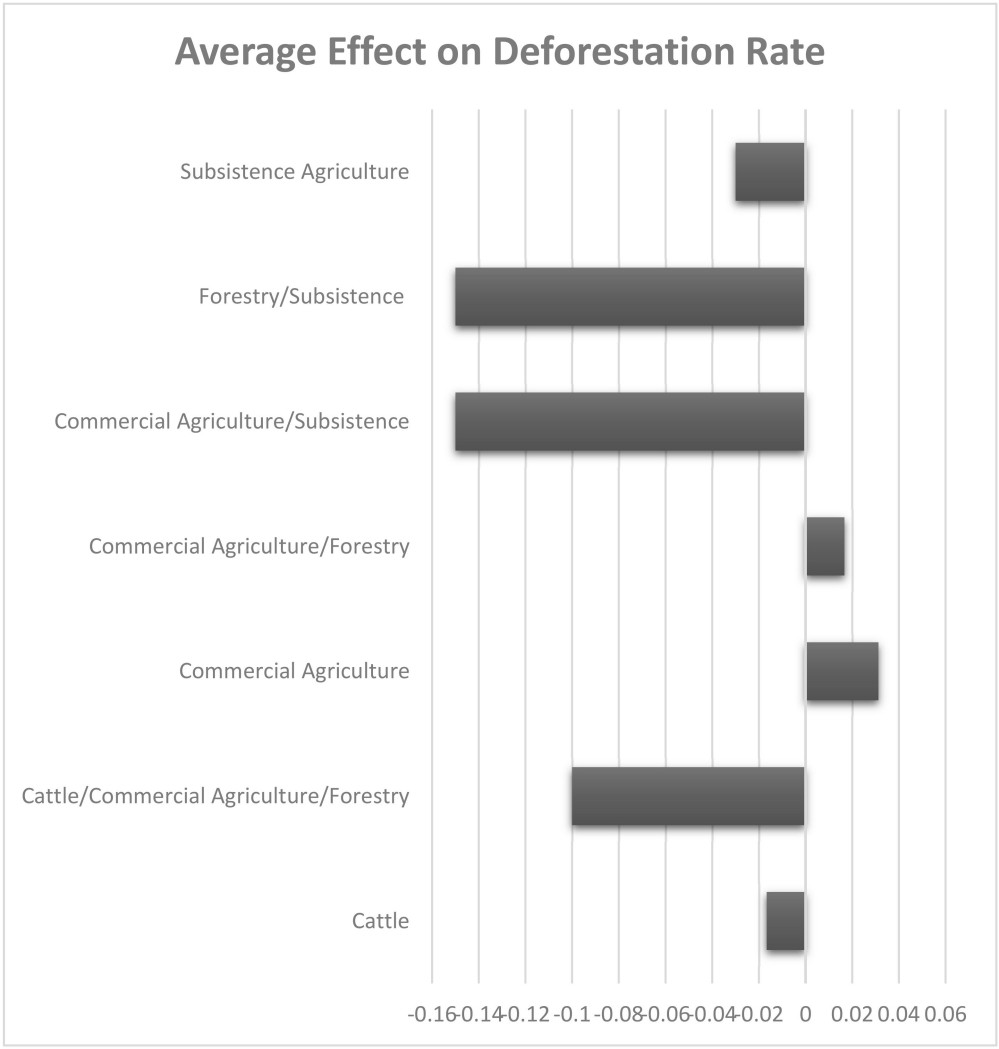

**Figure 5.** Average REDD+ effect on municipal deforestation rate according to major land uses.

**Table 4.** Average REDD+ effect on the municipal deforestation rate as per intervening funders and number of associated projects (CONAFOR = National Forestry Commission, UNDP = United Nations Development Program, AMREDD = Mexico REDD+ Alliance).

| Funders | Municipalities | Projects | Average Effect |
|---|---|---|---|
| AMREDD | 1 | 5 | 0.5 |
| AMREDD, UNDP | 4 | 17 | −0.1 |
| CONAFOR, AMREDD | 2 | 82 | −0.2 |
| CONAFOR, AMREDD, UNDP | 6 | 363 | 0.03 |
| CONAFOR, UNDP | 2 | 31 | −0.02 |
| UNDP | 24 | 47 | −0.07 |

### 3.2. Community (Micro) Scale Analyses

Community or ejido-scale DID regression results were similar to the municipal-scale results, showing no significant effect of the interaction term (DID = Time × REDD, $p$ = 0.738). The overall regression model, while significant ($p$ = 0.0001), had a poor fit ($R^2$ = 0.02). Significant covariates associated with forest cover loss were the ejido area parceled as private land (PrivParc, $p$ = 0.0005) and the area under agricultural or cattle production (CatpastPer, $p$ = 0.0001). The variability in proportion of forest loss rates (0–8%) among community sample units was large compared to municipal sample units. Mean annual rates of forest cover change before REDD+ interventions were 4.16 (SD = 22.1) in non-intervened ejidos and 1.95 (SD = 3.67) in REDD+ intervened ejidos. After REDD+ implementation, mean annual forest cover loss rates were reduced to 2.91 (SD = 14.67) in non-intervened ejidos and to 1.45 (SD = 2.68) in REDD+ intervened communities. Mean changes in annual forest cover rate after REDD+ intervention were −1.23 (SD = 11.07) in non-intervened ejidos and −0.50 (SD = 1.60) in ejidos with REDD+ projects. Matched control units (non REDD+ communities) derived from PSM results reduced variability and had a much closer mean rate of forest cover change (−0.61, SD = 2.0) to that of REDD+ intervened communities (−0.45, SD = 1.45) (Figure 6). The difference was not strongly significant ($U$ = 11399.5, $p$ = 0.09) but significant to the $\alpha$ = 0.10 level (Figure 7). Although, forest cover loss decreased in REDD+ -intervened ejidos, the reduction was less than in control ejidos, showing overall REDD+ ineffectiveness in community territories on the Yucatan Peninsula.

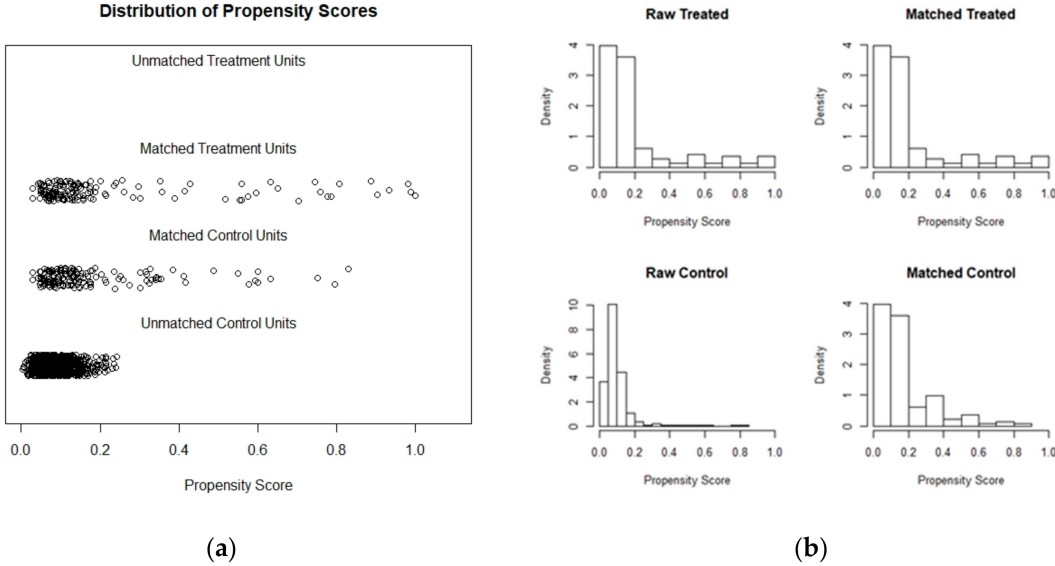

(**a**)                                        (**b**)

**Figure 6.** PSM results showing matched treated (REDD+) and unmatched control ejidos or units (**a**) and distribution of raw and matched treated and control ejidos or units (**b**).

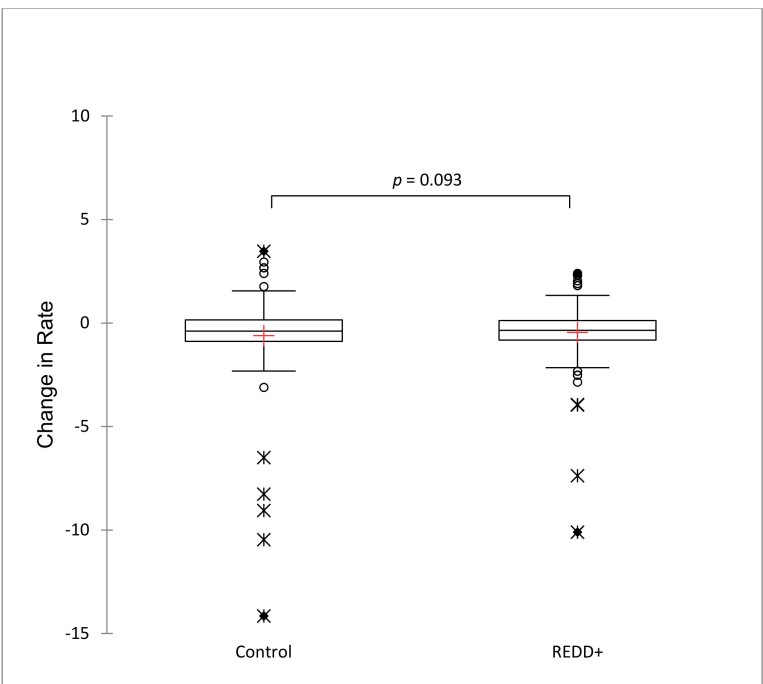

**Figure 7.** Boxplots and Mann–Whitney U two-sample comparison of change in annual forest cover loss after REDD+ implementation (2010–2018) in matched control communities (ejidos) derived from propensity score matching (0) and treated or REDD+ -intervened municipalities (1).

However, as in the meso-scale analysis, SCM outputs show mixed results when assessing REDD+ effectiveness at the community scale. Figure 4 maps the location of REDD+ -intervened communities based on effectiveness in reducing deforestation. In the state of Campeche, from a total of 37 ejidos with REDD+ -related projects (Table A1, Appendix A), half (19) did not have a reduction in forest loss rate from REDD+ intervention and the other half (18) demonstrated REDD+ effectiveness (Figure 8). Similarly, in Yucatan state, 33 communities (48%) indicated REDD+ effectiveness in reducing deforestation while 36 communities (52%) did not (Figure 8, Table A2). Quintana Roo had a greater proportion of REDD+ effective communities, 31 or 63%, compared to 18 or 37% that had constant or increased deforestation rates (Figure 8 and Table A3). Like the meso-scale SCM results, a clearer pattern is seen of communities that lacked REDD+ effectiveness in reducing deforestation clustered in southern municipalities of the Yucatan Peninsula bordering Belize and Guatemala. Table 5 shows the average effect on the rate of forest cover loss among communities grouped according to funders and number of projects. As observed at the municipal scale, the level of effort related to funders and number of projects fails to show any pattern related to REDD+ effectiveness. Contrary to meso-scale results, the nine communities that showed the greatest average reduction in rates of forest cover loss (−0.44) had projects (70) with all three major funders. For the most part, these were community forest ejidos located in the central municipality of Felipe Carrillo Puerto in Quintana Roo (Figure 4). A group of nine communities with a total of 40 projects funded by both CONAFOR and UNDP had the highest gain in forest cover loss and included large forestry ejidos in southern Quintana Roo, for example, Laguna Om, Caoba, and Tres Garantias. The level of effort in the latter two communities had been significant due to their Forest Stewardship Council (FSC) certification for sustainable forest management.

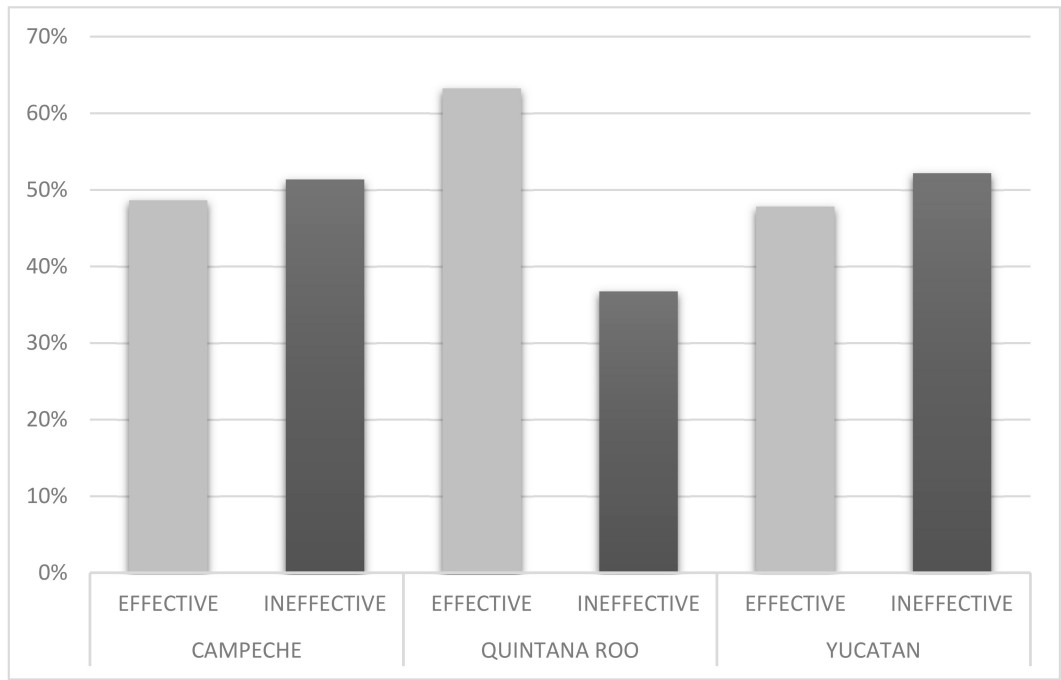

**Figure 8.** Proportions of REDD+ -intervened communities reducing deforestation rates (effective) or maintaining or increasing deforestation rates (ineffective) in Campeche, Quintana Roo, and Yucatan states.

**Table 5.** Average REDD+ effect on community (ejido) deforestation rate as per intervening funders and number of associated projects (CONAFOR = National Forestry Commission, UNDP = United Nations Development Program, AMREDD = Mexico REDD+ Alliance).

| Funders | Communities | Projects | Average Effect |
|---|---|---|---|
| AMREDD | 26 | 26 | −0.34 |
| CONAFOR | 35 | 125 | −0.26 |
| CONAFOR, AMREDD | 15 | 44 | −0.37 |
| CONAFOR, UNDP | 7 | 40 | 1.03 |
| CONAFOR, UNDP, AMREDD | 9 | 70 | −0.44 |
| UNDP | 51 | 58 | −0.39 |
| UNDP, AMREDD | 3 | 6 | 0.60 |

SCM community scale results show a more straightforward trend than the meso-scale evaluation on the role dominant land uses or production activities have on REDD+ outcomes (Figure 9). REDD+ effectiveness in reducing forest cover loss rates was most successful among communities where forestry and commercial or subsistence agriculture land uses are shared. Moreover, communities where subsistence agriculture or both commercial and subsistence agriculture are present were also effective in reducing the rate of forest loss. Contrarily, communities that failed to reduce deforestation and had the highest increase in deforestation were associated with regions where cattle ranching dominates or where cattle ranching, commercial agriculture, and forestry are combined. In general, where land use for cattle is present, REDD+ interventions tend to be ineffective; communities where commercial agriculture dominates also showed increased deforestation.

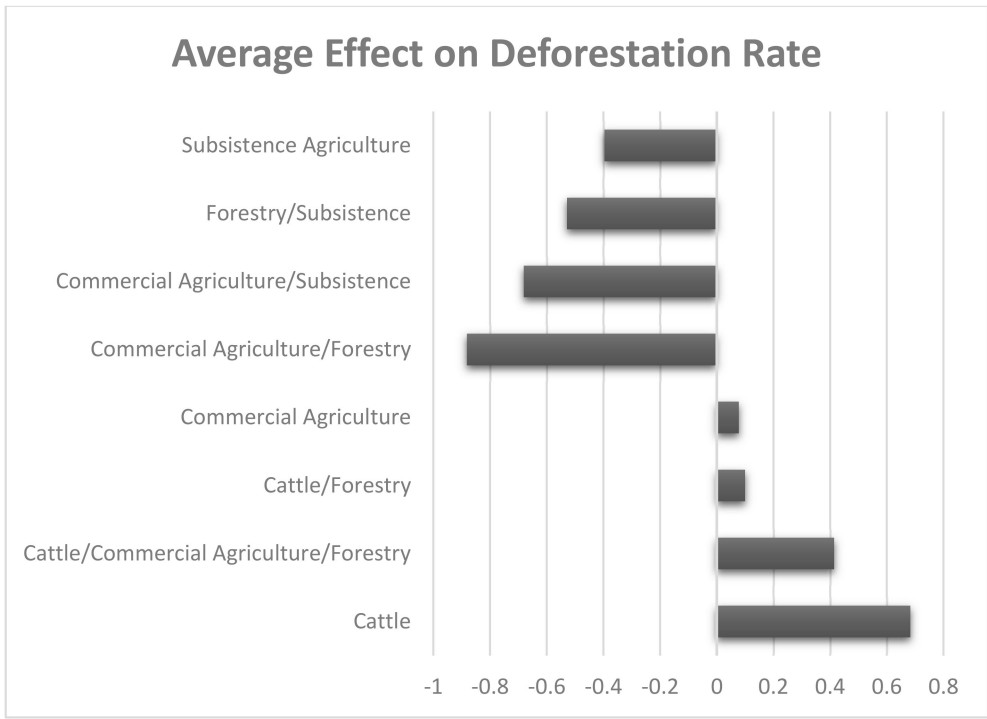

**Figure 9.** Average REDD+ effect on community (ejido) deforestation rate according to major land uses.

Figure 4 shows most of the REDD+ project communities in the state of Campeche without a reduction in forest cover loss located in the southern municipalities of Calakmul, Carmen, and Escarcega where cattle ranching is prominent. Fewer were located towards the central region of the state, Champoton, Hopelchen, and Calkini, where greater forest production and subsistence agriculture exist. Positive results for REDD+ communities in the municipality of Hopelchen are encouraging, being a deforestation hotspot. In addition, important forestry ejidos of Campeche state, for example, Nuevo Becal and Miguel Colorado in Calakmul and Champoton had positive REDD+ outcomes. Successes in the municipality of Calakmul are also meaningful to REDD+ goals. REDD+ effective cases in major commercial agriculture and cattle ranching regions of Campeche should be explored in detail to identify key institutional, governance, or socioeconomic factors that lead to REDD+ effectiveness.

Likewise, in Quintana Roo, REDD+ effectiveness is concentrated in communities of the central municipalities of Felipe Carrillo Puerto and Jose Maria Morelos where forest production is shared with subsistence or commercial agriculture. Felipe Carrillo Puerto is recognized for its low deforestation rate. On the other hand, communities in the southern municipality of Othon P. Blanco where cattle ranching, commercial agriculture, and forestry activities are present had an increase in deforestation. Contrary to the municipalities of Felipe Carrillo Puerto and José Maria Morelos, important forest management ejidos in the municipality of Othon P. Blanco, such as Botes, Caobas, Guadalajara, and Tres Garantías, were among communities that did not show reduced forest cover loss from REDD+ intervention. In these ejidos, cattle ranchings and sugarcane production are also common land uses. Other community cases showing increased forest loss in Quintana Roo included tourism-related communities in the municipality of Tulum (Chancen Chico and Hodzonot) and Lazaro Cardenas (Holbox).

In the state of Yucatan, ejidos that increased their deforestation despite REDD+ intervention were mostly located in the southern municipality of Tekax where commercial agriculture is shared with forestry activities and in the northern cattle ranching municipality of Tizimin. However, REDD+ effectiveness is noted in ejidos in eastern (Tzucacab, Tixcalpupul and Yaxcaba) and northern (Merida, Chocholá, and Tekal de Venegas) municipalities, many of these showing high deforestation rates before REDD+ intervention. Ejidos involved in forest management are few in the state of Yucatan, and most

REDD+ effective ejidos are dependent on small-scale commercial maize production and subsistence agriculture with larger areas under forest cover compared to other communities.

## 4. Discussion

Our results showed no overall effect of REDD+ interventions after 2010, deterring deforestation on the Yucatan Peninsula, concurring with a subnational assessment [7], trends documented in other countries [10], and a range of other forest conservation interventions across the globe [79]. At the municipal scale (meso-scale), the DID regression model signaled population density, agricultural immigration, fire, and urbanization as major determinants of forest cover loss, potentially overwhelming REDD+ interventions. At the community scale (micro-scale), DID regression indicated that land privatization and the area under cattle and agricultural production were associated with higher rates of forest loss, demonstrating local barriers to REDD+ effectiveness. This general trend in REDD+ effectiveness across the Yucatan Peninsula was strongly confirmed through our PSM analyses that revealed similar values in forest cover loss rates among control and REDD+ -intervened municipalities and communities before and after intervention, having no significant differences when statistically compared.

The generalized negative outcome of REDD+ interventions in municipalities and communities stresses the importance of improving subnational jurisdictional efforts and strategies to halt forest cover loss by strengthening state and municipal governments. Stickler et al. [7] also found that the three states of Campeche, Quintana Roo, and Yucatan failed to make any significant progress towards achieving their committed targets of reducing deforestation by 80% by 2020, accorded through their signing of the Rio Branco Declaration; the authors claim that lack of authority and limitations in subnational jurisdictions (state and municipal) and a centralized federal policy and approach towards REDD+ implementation are chiefly responsible. Bos et al. [10] hypothesized other potential reasons behind REDD+´s apparent lack of effectiveness in 23 subnational initiatives in Brazil, Peru, Cameroon, Tanzania, Indonesia, and Vietnam. Possible causes included flaws in the design of interventions and problems with assessment methods. In Mexico, the main design flaw identified in REDD+ interventions is not targeting important forest loss drivers (e.g., commercial agricultural and cattle land uses) while focusing on low deforestation land uses such as forest management and traditional agricultural practices [10,26,41]. The limitations in analysis methods pertain to differences in REDD+ projects and implementation periods and an emphasis on measuring deforestation outcomes without considering degradation, carbon storage, or accumulation. This study also has similar limitations in forest cover data and criteria used in the analyses that may influence generalized results of REDD+ ineffectiveness. For example, the time that the intervention begins in a community was not considered in our study. REDD+ in Mexico is only now phasing out of their preparation and piloting phases, so any effectiveness may be still too soon to detect.

Nevertheless, SCM applied in this research provided an additional tool that indicated REDD+ interventions had mixed effects on deforestation among intervened municipalities and ejido communities of the Yucatan Peninsula. Moreover, SCM proved to be a much more valuable means to evaluate REDD+ effectiveness, allowing for more useful assessments of individual cases of intervened municipalities or communities. We find that intervened municipalities with cattle and commercial agriculture (e.g., sugarcane, maize, and soy) land uses were typically unsuccessful in reducing forest cover loss. SCM results show that this trend is even stronger at the community scale. Ejidos where land use includes cattle rearing tend to increase their rates of forest loss the most, even when forestry is present. Communities where the dominant land use is mechanized farming also increase their loss of forest cover. In addition to land use, a pattern in the location of intervened municipalities and communities with a lack of REDD+ effectiveness was observed. REDD+ -intervened communities that substantially increased forest loss were concentrated in the southern peninsular region. This border zone with Guatemala and Belize is a remote forested frontier region where significant deforestation has occurred in the last 10 years due to the expansion of cattle pasture and agriculture, specifically in the

municipalities of Carmen, Candelaria, Calakmul, and Othon P. Blanco [35,80]. The area is also known for illicit activities of logging, drug smuggling, migration, and contraband which can interfere with the implementation of programs and projects. In neighboring Guatemalan forests, Devine et al. [80] describe how narcotrafficking is linked to cattle production and deforestation. There is little reason to doubt that this process may be occurring within Mexican forests near the border.

In addition, through SCM analyses, we were able to determine that the level of effort of REDD+ projects or funders working in a municipality or community had no bearing on effectiveness in reducing forest cover loss, leaving effectiveness closely tied to productive activities and location. For example, more funding and projects focused on the Calakmul Biosphere Reserve did not result in REDD+ effectiveness. However, some municipalities and communities in deforestation hotspots had improved outcomes from REDD+ interventions, including forest management, agricultural, and capacity-building projects. For example, despite the presence of suitable soils for mechanized agriculture in central municipalities of Hopelchen in Campeche and José María Morelos in Quintana Roo, a group of REDD+ -intervened ejidos reduced their forest cover loss. Among them were communities with forest management or agricultural communities with AMREDD projects aimed at strengthening governance and capacity-building. UNDP community-based small grant projects, particularly in agricultural areas of Yucatan state, also tended to reflect positive REDD+ outcomes. Other studies have also indicated that, when REDD+ is evaluated at the regional or meso scales the effectiveness in reducing deforestation is often not observed; on the other hand, effectiveness at local or micro scales is demonstrated in cases [10,15]. In that respect, our study also confirms the importance of evaluating REDD+ effectiveness at the local scale or project levels, using counterfactuals and quasi-experimental methods such as BACI and SCM [6,10,12,15]. Moreover, our results also corroborate that local funding and community-based projects may be more effective in reducing deforestation than regional jurisdictional interventions with government institutions [15,18,41].

With some caveats, our results echo the Skutsch and Turnhout [41] analysis which considers that REDD+ is not effective at tackling the most important drivers of deforestation and forest degradation. In the region, REDD+ is more often effective in locales with already low deforestation rates, particularly in those with community forest management and subsistence agriculture, where much of the REDD+ effort has concentrated [18]. Conversely, our results suggest that REDD+ is more often ineffective in communities with significant presence of industrial agriculture and cattle ranching where opportunity costs may be higher and low emissions development initiatives compete with business-as-usual incentives [44], illegal activities [80], and disproportionately larger agricultural subsidies [18]. Among the first lessons from initial REDD+ activities was the need to consider cross-sectoral transformation to change the course of drivers of deforestation and degradation [81]. Our study confirms that this challenge is still relevant in Mexico, especially the long-recognized need for better coordination between forest and agricultural policy. Some authors claim that REDD+ should be coupled with specific agricultural interventions (e.g., deforestation-free supply chain) [82].

Although important forestry communities with competing agricultural and cattle lands uses (e.g., Tres Garantias, Caobas, and Botes) showed no effects of REDD+ reducing deforestation. These ejidos were also located in the southern border zone region known to be affected by internal conflicts and illegal activities. However, our results show cases of forestry communities where commercial agriculture is present and REDD+ interventions were effective in reducing deforestation. We hypothesize that strengthening community governance and organizational capacity results in REDD+ effectiveness. However, the observed successes in communities certainly merit closer examination and further research for REDD+ planning and strategies. As Duchelle et al. [83] points out: "REDD+ on the ground is a customized basket of integrated interventions, including information, institutions, and incentives. Unsurprisingly, its effects are thus highly diverse."

Mixed effectiveness of REDD+ has been described in other evaluations, also showing reduced forest loss in communities with forestry and subsistence livelihoods and increased deforestation with cattle and commercial agriculture land uses [6,7,12,18,41,84]. Our study suggests that additional

factors, such as land tenure, local governance and organization, and financial credit and subsidies may also be driving heterogeneity among ejidos and suggest that future analysis include more details that can allow for a more nuanced and situated interpretation. Secure land tenure is often touted as the single most important factor for the success of conservation and development interventions such as REDD+ [85]. Land tenure rights in Mexican ejidos are relatively secure, and illegal land uses are limited compared to other regions of Latin America [86,87]. Our results highlight the insufficiency in secure land rights for ensuring REDD+ effectiveness. However, our results do show an important role of the type of land tenure in deforestation. At the micro-scale, forest cover loss is higher in privatized ejido lands. Notably, in community forestry ejidos, land tend tenure and resource use decisions tend to be collective, whereas for agriculture, individual land management and decisions prevail [87,88]. In that respect, community governance and organizational strength may also play a significant role in determining REDD+ effectiveness. This research shows cases of well-organized community forestry ejidos certified by Forest Steward Council for sustainable management (e.g., Nuevo Becal, Noh Bec, and Petcacab) with REDD+ effectiveness. Ramón Corona in Campeche state is among the few communities that have collectively implemented and participated in pilot silvopastoral and conservation agriculture projects, demonstrating REDD+ success in a high deforestation municipality [89]. The potential association between local REDD+ participation and legitimacy with the effectiveness of interventions [18,90] merit detailed exploration beyond individual case studies. REDD+ is being conceptually contested [91], and the governance processes associated with it are intensively scrutinized and sometimes challenged [3,18,88,92,93]. As the mixed effectiveness of REDD+ continues to be documented, the need to reexamine both the reaches and the blind spots of the current model of global sustainable forest governance become increasingly apparent [94].

Beyond how correct the theory of change is assumed by individual REDD+ projects in a specific locale [3], the opportunity costs associated with reducing deforestation and forest degradation can be expected to vary widely across sites, influencing intervention outcomes. Within the complex suite of factors that may explain the limited efficacy of REDD+ lies a relative meager funding [79]. International funding in Mexico has been very limited and decreasing compared with other countries like Brazil and Indonesia, where higher REDD+ effectiveness has been reported [7]. Moreover, coordinating interests is a major challenge for REDD+ as conflicting subsidies and policies pervade across REDD+ countries [3]. As occurs elsewhere [79], REDD+ projects often are a form of continuation of previously existing conservation and development initiatives, government programs, and governance platforms, now branded under the REDD+ umbrella (e.g., CONAFOR implementation of REDD+), while other REDD+ projects do represent an actual innovation for the region (e.g., AMREDD). Moreover, as various government-funded programs that historically supported CFM are defunded or cancelled, REDD+ has become a key element in community forestry policy. As pointed out by Pirard et al. [95], these compounded interventions over time often make the effects of individual initiatives "difficult to disentangle".

## 5. Conclusions

Though on average REDD+ -intervened municipalities and communities on the Yucatan Peninsula fail to demonstrate reduced forest cover loss, evaluation of individual cases reveals potential REDD+ successes. A tendency for low deforestation and community forest management have helped produce positive REDD+ outcomes in some cases. However, national market and policy incentives for crop and cattle production impede effectiveness in other cases. Strengthening community governance and organization and increased funding at the local level can help increase positive REDD+ results at the community and municipal scales. Community monitoring systems of forest cover and condition should be established in REDD+ -intervened ejidos to rigorously evaluate reductions in deforestation and degradation and enhancements in forest carbon stocks. Moreover, project evaluations should consider developing and integrating performance indicators such as of REDD+ effort (number of projects, funding, and period of intervention) and strength of community governance and legitimacy.

The general lack of effectiveness of REDD+ to tackle agriculture and cattle ranching as deforestation drivers highlights the need to research "commodity chain interventions to mitigate deforestation and land degradation" [96]. Emerging elements are expected to influence future land system dynamics in the region. New large mega-projects, such as the Tren Maya (the Mayan Train) that imply the construction of a touristic railroad that circuits across the entire Yucatan Peninsula, are likely to significantly impact regional socio-ecological systems [97] and would require close monitoring.

**Author Contributions:** Conceptualization, E.A.E.; methodology, E.A.E.; software, E.A.E.; validation, E.A.E. and J.A.S.-H.; formal analysis, E.A.E.; investigation, E.A.E. and J.A.S.-H.; resources, E.A.E. and J.A.S.-H.; writing—original draft preparation, E.A.E. and J.A.S.-H.; writing—review and editing, E.A.E., J.A.S.-H., C.L.B., G.C.O.C., and C.R.C. All authors have read and agreed to the published version of the manuscript.

**Funding:** This research received no external funding.

**Acknowledgments:** We thank Samaria Armenta Montero for assistance in data preparation.

**Conflicts of Interest:** The authors declare no conflict of interest.

## Appendix A

**Table A1.** Synthetic Control Method (SCM) results of treated unit effects for each REDD+ community (ejido) in the state of Campeche (CONAFOR = National Forestry Commission, UNDP = United Nations Development Program, AMREDD = Mexico REDD+ Alliance).

| Ejido | Municipality | Effect | REDD+ Intervention |
|---|---|---|---|
| 20 de Noviembre | Calakmul | 0.2 | CONAFOR, UNDP, AMREDD |
| Constitución | Calakmul | 0.2 | CONAFOR |
| Gral. Alvaro Obregon | Calakmul | 1.5 | CONAFOR, AMREDD |
| La Lucha | Calakmul | −2.5 | AMREDD |
| N.P. Ricardo Payro | Calakmul | 0 | AMREDD |
| N.P. Santa Rosa | Calakmul | 2 | CONAFOR, AMREDD |
| Nuevo Becal | Calakmul | −0.3 | CONAFOR, UNDP, AMREDD |
| Nuevo Progreso | Calakmul | −4 | AMREDD |
| Xbonil | Calakmul | 0.5 | CONAFOR |
| Becal | Calkini | −1.8 | UNDP |
| Chun Ek | Calkini | 2.5 | CONAFOR, AMREDD |
| Chunhuas | Calkini | 0 | CONAFOR, UNDP |
| Chunyaxnic | Calkini | −0.7 | CONAFOR |
| Hampolol | Campeche | −3.2 | UNDP |
| Atasta | Carmen | 0.7 | UNDP |
| Chekubul | Carmen | 0.25 | UNDP |
| Los Manantiales | Carmen | 1.3 | AMREDD |
| Sabancuy | Carmen | −6 | UNDP |
| San Ant. Cardenas | Carmen | 0.7 | UNDP |
| Champoton | Champoton | 1.75 | CONAFOR |
| Felipe Carrillo Puerto | Champoton | 0 | UNDP |
| Kilometro 67 | Champoton | −0.3 | CONAFOR |
| Lazaro Cárdenas | Champoton | −5 | CONAFOR |
| Miguel Colorado | Champoton | −0.5 | CONAFOR, AMREDD |
| Silvituc | Escarcega | 0 | CONAFOR, AMREDD |
| El Lechugal | Escárcega | −0.3 | CONAFOR, AMREDD |
| El Manantial | Escárcega | −2.5 | AMREDD |
| Matamorros | Escárcega | 0 | CONAFOR |
| N.P. Altamira Zinapro | Escárcega | 0.7 | AMREDD |
| Cancabchen | Hopelchen | −0.5 | CONAFOR |
| Hopelchen | Hopelchen | 0.2 | AMREDD |
| Ich Ek | Hopelchen | −1 | CONAFOR, AMREDD |
| Ramon Corona | Hopelchen | −6 | CONAFOR, AMREDD |
| San Fran.Suc Tuc | Hopelchen | 0.5 | CONAFOR, AMREDD |
| Xmaben | Hopelchen | −0.3 | CONAFOR |
| Xmejia | Hopelchen | −1.5 | CONAFOR |

**Table A2.** Synthetic Control Method (SCM) results of treated unit effects for each REDD+ community (ejido) in the state of Yucatan (CONAFOR = National Forestry Commission, UNDP = United Nations Development Program, AMREDD = Mexico REDD+ Alliance).

| Ejido | Municipality | Effect | REDD+ Intervention |
|---|---|---|---|
| Alfonso Caso | Tekax | −1.5 | AMREDD |
| Alfonso Caso II | Tekax | 1 | UNDP, AMREDD |
| Bekanchen | Tekax | 0.1 | AMREDD |
| Cantamayec | Cantamayec | −2.5 | UNDP |
| Catmis | Tzucacab | −4 | AMREDD |
| Caucel | Merida | −0.2 | CONAFOR |
| Celestun | Celestun | 0.25 | UNDP |
| Chacsinkin | Chacskinkin | −0.5 | UNDP |
| Chochola | Chochola | −1 | UNDP |
| Cholul | Cantamayec | −0.5 | UNDP |
| Cholul | Merida | −0.5 | UNDP |
| Chuchub | Tixmehuac | −1.2 | UNDP |
| Chulutan | Vallalodid | −0.5 | AMREDD |
| Colonia Yucatan | Tizimin | 6 | AMREDD |
| Dzilam de Bravo | Dzilam de Bravo | 0 | UNDP |
| Dzoncauich | Dzoncauich | −1 | UNDP |
| Ekbalam | Tzucacab | −1.5 | AMREDD |
| Felipe Carrillo Puerto | Dzilam Gonzales | 40 | UNDP |
| Francisco Villa | Tizimin | 0.25 | AMREDD |
| Hocaba | Hocaba | 0.25 | UNDP |
| Huacpelchen | Huacpelchen | −0.2 | AMREDD |
| Izamal | Izamal | 0.75 | CONAFOR |
| Kimbila | Izamal | 0 | UNDP |
| Kimbila | Tixmehuac | 1 | UNDP |
| Kinil | Tekax | −0.75 | AMREDD |
| Mani | Mani | 1 | UNDP |
| Manuel Cepeda Peraza | Tizimin | −2.5 | AMREDD |
| Mayapan | Mayapan | −1.5 | UNDP |
| Molas | Merida | −0.25 | UNDP |
| NCPE Poboch Nuevo | Tekax | 0 | CONAFOR |
| NCPA San Agustin | Tekax | 1 | UNDP, AMREDD |
| NCPE San Salvador | Tekax | 0 | AMREDD |
| Nenela | Cantamayec | 0.5 | UNDP |
| Noh-Bec | Tzucacab | 2 | UNDP |
| Nohuayun | Tetiz | 0 | UNDP |
| Oxkutzcab | Oxkutzcab | 0.7 | UNDP |
| Petac | Merida | −1.25 | UNDP |
| Petecbiltun | Uman | 2 | UNDP |
| Sabacche | Techo | −1.5 | UNDP |
| Sabacche | Tixmehuac | 0 | UNDP |
| San Antonio Mulix | Uman | 0 | UNDP |
| San Crisanto | Sinanche | 0 | UNDP |
| San Jorge | Tixkokob | −2.5 | CONAFOR |
| San Marcos | Tekax | −0.5 | UNDP, AMREDD |
| San Marcos | Yaxcaba | −0.5 | UNDP |
| San Salvador Piste | Tzucacab | −1.25 | UNDP |
| San Simon | Santa Elena | −0.5 | AMREDD |
| Sisal | Hunucma | 0.1 | UNDP |
| Sudzal | Sudzal | 0.2 | UNDP |
| Tanil | Uman | 0.2 | UNDP |
| Teabo | Teabo | 0 | UNDP |
| Tebec | Uman | −1.25 | UNDP |
| Tekal | Tekal de Venegas | −0.5 | UNDP |
| Tekax | Tekax | −2.5 | UNDP |
| Ticimul | Chankom | 0.5 | UNDP |

**Table A2.** *Cont.*

| Ejido | Municipality | Effect | REDD+ Intervention |
|---|---|---|---|
| Ticimul | Uman | 4 | UNDP |
| Tinuncah | Yaxcaba | −1.25 | AMREDD |
| Tixcacalcupul | Tixccacalcupul | −0.7 | UNDP |
| Tixmeuac | Tixmehuac | 0 | UNDP |
| Ucu | Ucu | 0.1 | UNDP |
| Xanaba | Izamal | 0.7 | UNDP |
| Xcanatun | Merida | 0 | UNDP |
| Xkalakyodzonot | Tizimin | 6 | AMREDD |
| Xul | Oxkutzcab | 1.5 | UNDP |
| Yaxcopoil | Uman | −0.3 | UNDP |
| Yaxhachen | Peto | 2 | UNDP |
| Yodzonot | Calotmul | −0.7 | UNDP |

**Table A3.** Synthetic Control Method (SCM) results of treated unit effects for each REDD+ community (ejido) in the state of Quintana Roo (CONAFOR = National Forestry Commission, UNDP = United Nations Development Program, AMREDD = Mexico REDD+ Alliance).

| Ejido | Municipality | Effect | REDD+ Intervention |
|---|---|---|---|
| Bacalar | Bacalar | 0 | UNDP |
| Chacchoben | Bacalar | −2 | CONAFOR |
| Alvaro Obregón | Bacalar | 3 | UNDP |
| X-Hazil Norte | Felipe C, Puerto | 2 | AMREDD |
| Andrés Quintana Roo | Felipe C. Puerto | 0.2 | CONAFOR |
| Betania | Felipe C. Puerto | −1 | CONAFOR, UNDP, AMREDD |
| Chancah Derrepente | Felipe C. Puerto | −1.7 | CONAFOR |
| Chunhuas | Felipe C. Puerto | −0.7 | CONAFOR, UNDP, AMREDD |
| Chunyaxche | Felipe C. Puerto | −0.5 | UNDP |
| Cuauhtemoc | Felipe C. Puerto | −2 | CONAFOR, AMREDD |
| Dzula | Felipe C. Puerto | −1 | CONAFOR, UNDP, AMREDD |
| Felipe C. Puerto | Felipe C. Puerto | 0.5 | CONAFOR, UNDP, AMREDD |
| Laguna Kana | Felipe C. Puerto | −0.2 | CONAFOR, UNDP, AMREDD |
| NCPE Gral. E. Zapata | Felipe C. Puerto | 2 | CONAFOR |
| Naranjal Poniente | Felipe C. Puerto | −0.7 | CONAFOR |
| Noh Bec | Felipe C. Puerto | −1.3 | CONAFOR, UNDP, AMREDD |
| Petcacab | Felipe C. Puerto | −0.5 | CONAFOR |
| San Francisco Ake | Felipe C. Puerto | −4.5 | CONAFOR |
| Santa Maria Poniente | Felipe C. Puerto | −0.8 | CONAFOR |
| Tabi | Felipe C. Puerto | −0.5 | CONAFOR |
| X-Hazil | Felipe C. Puerto | −0.2 | UNDP, AMREDD |
| X-Yatil | Felipe C. Puerto | −1 | CONAFOR |
| Yoactun | Felipe C. Puerto | −0.5 | CONAFOR, AMREDD |
| Gavilanes | Jose M. Morelos | −2.5 | CONAFOR |
| Javier Rojo Gomez | Jose M. Morelos | −0.5 | CONAFOR |
| Plan de la Noria Oriente | Jose M. Morelos | 0 | CONAFOR |
| Pozo Pirata | Jose M. Morelos | −2.5 | CONAFOR |
| Puerto Arturo | Jose M. Morelos | −2 | CONAFOR, UNDP, AMREDD |
| San Antonio Tuk | Jose M. Morelos | 0 | CONAFOR, UNDP |
| Emiliano Zapata | José M. Morelos | 2 | CONAFOR, AMREDD |
| La Esperanza | José M. Morelos | −2 | CONAFOR, AMREDD |
| Rancho Viejo | José M. Morelos | −0.5 | AMREDD |
| San Cristobal | José M. Morelos | −1.3 | CONAFOR, AMREDD |
| San Felipe III | José M. Morelos | −0.5 | AMREDD |
| San Marcos | José M. Morelos | −2.5 | UNDP |
| Tabasco | José M. Morelos | −0.5 | CONAFOR, AMREDD |
| Venustiano Carranza | José M. Morelos | −0.8 | CONAFOR |

**Table A3.** *Cont.*

| Ejido | Municipality | Effect | REDD+ Intervention |
|---|---|---|---|
| Solferino | Lazaro Cardenas | −0.2 | UNDP |
| Holbox | Lázaro Cardenas | 3 | UNDP |
| El Cafetal | Othon P. Blanco | −0.2 | CONAFOR |
| Guadalajara | Othon P. Blanco | 3 | CONAFOR |
| Los Divorciados | Othon P. Blanco | 0.5 | CONAFOR |
| Tres Garantías | Othon P. Blanco | 2 | CONAFOR, UNDP |
| Botes | Othón P. Blanco | 6 | CONAFOR |
| Caobas | Othón P. Blanco | 3 | CONAFOR, UNDP |
| Laguna Om | Othón P. Blanco | 1 | CONAFOR, UNDP |
| Chanchen Chico | Tulum | 0.5 | CONAFOR, UNDP |
| Hodzonot | Tulum | 0 | CONAFOR |

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
