# Peer review of "Mixed Effectiveness of REDD+ Subnational Initiatives after 10 Years of Interventions on the Yucatan Peninsula, Mexico"

_forests, doi:10.3390/f11091005_

Round 1
Reviewer 1 Report
Broad comments highlighting areas of strength and weakness.
The strengths of the document include the number of case sites that have been able to be examined and over a large area. Insights are gained into the overall success of the interventions in a very important forested region of the world. Overall I enjoyed reading this paper and also well aware of the challenge in conducting such as study. But think this paper would benefit from clarification in certain parts. In addition more in the results and discussion to highlight the patterns between the interventions that have resulted in forest gain (and what we can learn from these), those that have seen no change (also a positive result depending on the projects aim and the forest cover status to begin with) vs. those with ongoing forest loss.
The weaknesses include:
INTRODUCTION
- In the introduction suggest clarifying the whole section that describes budgets line 85 - 106. This was so dense and hard to read. The sentence from 85 – 88 is too cluttered. Also could you also indicate how funds have decreased since 2014 i.e. by how much? What are possible reasons for this?
- Suggest adding a table to summarise the numbers for each type of donor and what the funds go towards. This will also help later when you refer to the REDD+ interventions but it is unclear what this means in practice.
- UNDP
- CONFAFOR, UNDPE, AMREDD
- AMREDD
- ETC
if this does not tell us much about the intention itself by referring to those funders in those location than perhaps there’s less need to put emphasis there later on…you just need to flag that who they are funder's and just describe them as REDD+ interventions. or can we catergorized in another way e.g. types of REDD+ land management activities?
- For the section on ‘1.2. Challenges to REDD+ Implementation and Evaluation’ suggest to add a map - as for readers not familiar with the region the names of the various regions and sites gets confusing. This may help to clarify if you’re also talking about the whole area including Belize and northern Guatemala or already focused on Mexico in your descriptions. Suggest you comment on whether in the neighbouring countries REDD+ interventions are also being implemented/ forest cover dynamics etc. Also this may help to justify why you focus on the Yucatan peninsula area of the Selva Maya specifically.
MATERIALS AND METHODS
- Figure 1 (the map) is unclear but very useful. This map should be improved to indicate more clearly where the interventions are and forest cover status. Currently the map is too cluttered so suggest to use two smaller maps 1. Interventions and the other 2. forest cover loss/tree. For tree cover perhaps use a stronger colour gradient to indicate the high and low tree cover %. Also the municipalities’ name labels need to be clearer.
RESULTS
- Try to say more on the interventions themselves and the impacts. You distinguish between the types of donor interventions (e.g. UNDP, CONAFOR AMREDD) but what is it about those interventions that make them more effective or not that can be explain the impacts. Is it the funders? Or is it down to individual projects?
- Suggest you comment on the timing of interventions as this is likely to have a big role (where they all get established around the same time?). Also what about the phases if so far effort shave been on planning and ‘Readiness’ phase of REDD+ this may not have results in concrete actions on the ground. These are some other types of explanations as to why effectiveness might not yet be evident to consider/note.
- In the results section the naming of every municipality and Community makes the section difficult to read and follow. Do all the readers need to know the name of every single place? Think it is more important here to flag the patterns between those sets of municipalities/communities and the number of municipalities within those sets are more telling then to explain their success / or in effectiveness. Also could ‘no change’ also be seen as a positive result e.g. its better than loss. It may be an area that had high forest cover to begin with and if this stayed intact this would be a positive result?
- Also suggest you cut down the tables rather than list all locations you could have tables that summarise the numbers here e.g. Number of municipalities / communities with no effect vs those with a positive effect (forest recovery) vs those with a negative effect (forest loss). The tables you have now could still go into an appendix e.g. table 3 /4/5
DISCUSSION
- Suggest to add the key figures from the findings e.g... X % REDD+ sites had decreases in deforestation etc at the municipality level, and x % at the community level etc.
- Currently there is an over emphasis on other studies - the results from this study need to be emphasized then related back to the supporting literature. Otherwise its not entirely clear what makes this study really stand out. This is a really important point as will justify the added value of this study.
APPENDIX
- Not convinced how helpful all the graphs that are currently in the appendix are?!
Specific comments referring to line numbers, tables or figures.
Line 62 and 140 - avoid the word implant – try and use implement
Line 87 – what’s an IRDB spell out in full?
Line 108 – 111 – shorten sentence
Point made in line 153 is repetitive of an earlier point
Line 47 – Reword sentence ‘was formally initiated in 2009…
Line 317 DID=data$time*data$redd. Is this written correctly?
Line 367 for: Differences in differences (or differences-in-differences) suggest writing it the same way and/or use the acronym – need to be consistent
Line 537 – REDD+ is not being effective – rephrase
Line 561 – ‘Notably, with 561 different shades’ - not sure what you mean here suggest rephrasing
Line 511- 516 sentence is too long
Line 462 – the suggestions to” studied and monitored for strategic planning” not sure this makes sense...
Line 103 swap word ‘current’ for ‘ongoing’
Line 106 delete ‘a sum of’
Line 132 rephrase ‘or consented’
Author Response
Thank you for your suggestions and corrections to improve this manuscript. Revisions are described below in italic.
INTRODUCTION
We eliminated up to 40 lines of text in the History of REDD+ in Mexico section of the manuscript, specifically pertaining to funding and other details in order to shorten and clarify the introduction for readers. In addition we added a short introductory section (lines 41-54) that sets the stage for the importance and purpose of the study.
- In the introduction suggest clarifying the whole section that describes budgets line 85 - 106. This was so dense and hard to read. The sentence from 85 – 88 is too cluttered. Also could you also indicate how funds have decreased since 2014 i.e. by how much? What are possible reasons for this?
These lines were removed, and description of funding and budgets was eliminated to shorten the History of REDD section as suggested by Academic editor and Reviewer.
- Suggest adding a table to summarise the numbers for each type of donor and what the funds go towards. This will also help later when you refer to the REDD+ interventions but it is unclear what this means in practice.
- UNDP
- CONFAFOR, UNDPE, AMREDD
- AMREDD
- ETC
if this does not tell us much about the intention itself by referring to those funders in those location than perhaps there’s less need to put emphasis there later on…you just need to flag that who they are funder's and just describe them as REDD+ interventions. or can we catergorized in another way e.g. types of REDD+ land management activities?
Tables 4 (Line 498) and 5 (Line 558) were added to Results summarizing funders and number of projects intervening in municipalities and communities and the associated average effect of REDD+ effectiveness in reducing deforestation. In addition, Figures 5 (Line 500) and 9 (Line 588) show REDD+ effectiveness summarized by land use practices in municipalities and communities, respectively.
- For the section on ‘1.2. Challenges to REDD+ Implementation and Evaluation’ suggest to add a map - as for readers not familiar with the region the names of the various regions and sites gets confusing. This may help to clarify if you’re also talking about the whole area including Belize and northern Guatemala or already focused on Mexico in your descriptions. Suggest you comment on whether in the neighbouring countries REDD+ interventions are also being implemented/ forest cover dynamics etc. Also, this may help to justify why you focus on the Yucatan peninsula area of the Selva Maya specifically.
We modified Figure 1 (Study area map, Line 214) and refer to it in the “Challenges to REDD+….” section, indicating the names of states, protected areas, etc. mentioned in the text. Figure 1 also includes the extent of forest cover of the Selva Maya in an insert and clarifies that our study area only includes the Mexico (states of Campeche, Quintana Roo and Yucatan). Availability and uniformity of data are the main reasons we focus on Mexico, although there are REDD+ interventions in Guatemala and recently in Belize. The border zone between Mexico and Central America is now mentioned in the text (Discussion) about the impacts illicit activities and migration may have on REDD+ effectiveness.
MATERIALS AND METHODS
- Figure 1 (the map) is unclear but very useful. This map should be improved to indicate more clearly where the interventions are and forest cover status. Currently the map is too cluttered so suggest to use two smaller maps 1. Interventions and the other 2. forest cover loss/tree. For tree cover perhaps use a stronger colour gradient to indicate the high and low tree cover %. Also the municipalities’ name labels need to be clearer.
We improved the map as described above and added a second map (Figure 4, Line 472) showing the distribution of REDD+ municipalities and communities according to the degree of REDD+ effectiveness in reducing forest cover loss. Classification and gradient of % forest cover was modified for visual clarity.
RESULTS
- Try to say more on the interventions themselves and the impacts. You distinguish between the types of donor interventions (e.g. UNDP, CONAFOR AMREDD) but what is it about those interventions that make them more effective or not that can be explain the impacts. Is it the funders? Or is it down to individual projects?
In the Results section of the manuscript we modified the text and presentation of the SCM results in order to clarify and improve the interpretation of trends and patterns observed among REDD+ effective on non-effective cases related to project effort and funders and dominant land use practices. An additional map (Figure 4, Line 472) was included to show the location and distribution of REDD+ intervened municipalities and communities according the degree of effectiveness in reducing rate of forest cover loss. A table was added (Table 4, Line 498) that shows average REDD+ effect according to Funders and number of project interventions, and a figure inserted (Figure 5, Line 500) summarizing the average effect based on land use. Additional text (Lines 476-497) was added describing observed trends and patterns at the meso-scale.
For SCM community-scale results, Lines 531-552 were integrated to describe the mixed effectiveness among communities in each state (Campeche, Quintana Roo and Yucatan) of the study area. These results are summarized in a new figure (Figure 8) added to the manuscript. The three long tables showing REDD+ effect by community in each state were removed from the text and moved to the Appendix. The SCM generated graphs in the Appendix of the previous version were removed. A table (Table 5, Line 558) showing the average REDD+ effect with respect to funders and projects and a figure (Figure 9, Line 588) showing average effect by land uses were added. Additional text (Lines 561-581) was written to describe the patterns and trends of REDD+ effectiveness or lack of. Specific examples of communities in each state are offered (Lines 592-633)
- Suggest you comment on the timing of interventions as this is likely to have a big role (where they all get established around the same time?). Also what about the phases if so far effort shave been on planning and ‘Readiness’ phase of REDD+ this may not have results in concrete actions on the ground. These are some other types of explanations as to why effectiveness might not yet be evident to consider/note.
We comment on timing of REDD+ interventions and phases in affecting overall effectiveness results.
- In the results section the naming of every municipality and Community makes the section difficult to read and follow. Do all the readers need to know the name of every single place? Think it is more important here to flag the patterns between those sets of municipalities/communities and the number of municipalities within those sets are more telling then to explain their success / or in effectiveness. Also could ‘no change’ also be seen as a positive result e.g. its better than loss. It may be an area that had high forest cover to begin with and if this stayed intact this would be a positive result?
The results were modified a described above to improve clarity and readability. We focus on patterns and trends related to funders, projects and land uses at municipal and community scales. We don´t include “No change” as effective since the effect shows if the rate of forest cover loss was reduced (negative sign) or increased (positive sign). A neutral sign would mean that the forest cover loss rate remained the same not forest cover area. However, we do make distinctions of these cases.
- Also suggest you cut down the tables rather than list all locations you could have tables that summarise the numbers here e.g. Number of municipalities / communities with no effect vs those with a positive effect (forest recovery) vs those with a negative effect (forest loss). The tables you have now could still go into an appendix e.g. table 3 /4/5
We removed the tables of results of communities and moved them to the appendix. Results are now modified and include additional figures and tables described above.
DISCUSSION
- Suggest to add the key figures from the findings e.g... X % REDD+ sites had decreases in deforestation etc at the municipality level, and x % at the community level etc.
This figure was added in results, Figure 8, Line 553
- Currently there is an over emphasis on other studies - the results from this study need to be emphasized then related back to the supporting literature. Otherwise its not entirely clear what makes this study really stand out. This is a really important point as will justify the added value of this study.
The Discussion was modified to emphasize the major take-home messages from the results that suggest why some municipalities and communities are effective and why other are not. New text highlighting and emphasizing our results are introduced in Lines 635-646 and Lines 671-701
APPENDIX
- Not convinced how helpful all the graphs that are currently in the appendix are?!
We removed these figures
Specific comments referring to line numbers, tables or figures.
Line 62 and 140 - avoid the word implant – try and use implement
Changed
Line 87 – what’s an IRDB spell out in full?
Deleted
Line 108 – 111 – shorten sentence
Sentence shortened
Point made in line 153 is repetitive of an earlier point
Deleted
Line 47 – Reword sentence ‘was formally initiated in 2009…
Deleted
Line 317 DID=data$time*data$redd. Is this written correctly?
Line 367 for: Differences in differences (or differences-in-differences) suggest writing it the same way and/or use the acronym – need to be consistent
Changed
Line 537 – REDD+ is not being effective – rephrase
Sentence edited.
Line 561 – ‘Notably, with 561 different shades’ - not sure what you mean here suggest rephrasing
We deleted that
Line 511- 516 sentence is too long
Sentence edited.
Line 462 – the suggestions to” studied and monitored for strategic planning” not sure this makes sense...
Sentence edited.
Line 103 swap word ‘current’ for ‘ongoing’
Changed
Line 106 delete ‘a sum of’
Deleted
Line 132 rephrase ‘or consented’
Rephrased
Reviewer 2 Report
The paper entitled as “Mixed effectiveness of REDD+ subnational initiatives after 10 years of interventions on the Yucatan Peninsula, Mexico” compiled long history of REDD+ and related activities in Mexico, deeply. Firstly, I could not identify the concept or idea of the study. Dose authors aimed to compile histories? I did not have any scientific outcome from this study, just received history of the REDD+ in Mexico. Considering journal’s objectives, I would give only simple comments as follows and decide this literature is to be rejected.
Specific comments:
- In the part of Introduction, the authors described detailed histories of REDD+ in Mexico. I would not understand the objectives of the paper. More concreate objectives should be mentioned. For example, pert of “1.3. Resarch Objetives and Methodological Approach” does not mentioned what authors aimed to clarify in this paper.
- Figure 1 is very low quality and I could not clarify land use/cover pattern in this map.
- Left side of the Figure 2 is hard to understand. What is points in the figure?
- As same as above, I could not understand meanings of the Figure 3. What is vertical axis label?
- As same as left side of Figure 2, I could not understand Figure 4 and Figure 5.
Author Response
Thank you for your suggestions and corrections to improve this manuscript. Revisions are described below.
INTRODUCTION
We eliminated up to 40 lines of text in the History of REDD+ in Mexico section of the manuscript, specifically pertaining to funding and other details in order to shorten and clarify the introduction for readers. In addition we added a short introductory section (lines 41-54) that sets the stage for the importance and purpose of the study.
RESULTS
In the Results section of the manuscript we modified the text and presentation of the SCM results in order to clarify and improve the interpretation of trends and patterns observed among REDD+ effective on non-effective cases related to project effort and funders and dominant land use practices. An additional map (Figure 4, Line 472) was included to show the location and distribution of REDD+ intervened municipalities and communities according the degree of effectiveness in reducing rate of forest cover loss. A table was added (Table 4, Line 498) that shows average REDD+ effect according to Funders and number of project interventions, and a figure inserted (Figure 5, Line 500) summarizing the average effect based on land use. Additional text (Lines 476-497) was added describing observed trends and patterns at the meso-scale.
For SCM community-scale results Line 531-552 were integrated to describe the mixed effectiveness among communities in each state (Campeche, Quintana Roo and Yucatan) of the study area. These results are summarized in a new figure (Figure 8, Line 553) added to the manuscript. The three long tables showing REDD+ effect by community in each state were removed from the text and moved to the Appendix. The SCM generated graphs in the Appendix of the previous version were removed. A table (Table 5, Line 558) showing the average REDD+ effect with respect to funders and projects and a figure (Figure 9, Line 588) showing average effect by land uses were added. Additional text (Lines 561-581) was written to describe the patterns and trends of REDD+ effectiveness or lack of. Specific examples of communities in each state are offered (Lines 592-633)
DISCUSSION
The Discussion was modified to emphasize the major take-home messages from the results that suggest why some municipalities and communities are effective and why other are not. New text highlighting and emphasizing our results are introduced in Lines 635-646 and Lines 671-701
GRAPHS AND FIGURES
We modified and clarified all graphs and figures mentioned in your review. See above.
Round 2
Reviewer 2 Report
thank you for revisions. the most of parts I pointed out were improved. I decided this manuscript is acceptable.
Author Response
Thank you for your review and suggestions to improve the manuscript. We are grateful that you find the modifications of our second revision suitable for publication. Regards.